# Micro-Macro Retrieval: Reducing Long-Form Hallucination in Large Language Models

**Yujie Feng**[1,2*]**, Jian Li**[1*]**, Zhihan Zhou**[3]**, Pengfei Xu**[1]**, Yujia Zhang**[1]
**Xiaoyu Li**[1]**, Xiaohui Zhou**[1]**, Alan Zhao**[1]**, Xi Chen**[1†]**, Xiao-Ming Wu**[2†]
[1]Solar System of OVB, Tencent, China
[2]The Hong Kong Polytechnic University, Hong Kong S.A.R. [3]Jilin University, China

## Abstract

Large Language Models (LLMs) achieve impressive performance across many tasks but remain prone to hallucination, especially in long-form generation where redundant retrieved contexts and lengthy reasoning chains amplify factual errors. Recent studies highlight a critical phenomenon: the closer key information appears to the model outputs, the higher the factual accuracy. However, existing retrieval-augmented language models (RALMs) lack effective mechanisms to ensure this proximity — external evidence is injected into reasoning via multi-turn retrieval, but this cannot ensure key information stays close to the outputs. We propose **Micro–Macro Retrieval** (**M**$^2$**R**), a novel *retrieve-while-generate* framework to fill this gap. At the macro level, M$^2$R retrieves coarse-grained evidence from external sources; at the micro level, it extracts essential results from a key information repository built during reasoning and reuses them while generating answers. This design directly addresses the key-information–to-output proximity bottleneck, effectively reducing hallucination in long-form tasks. M$^2$R is trained with a curriculum learning–based reinforcement learning strategy using customized rule-based rewards, enabling stable acquisition of retrieval and grounding skills. Extensive experiments across different benchmarks demonstrate the effectiveness of M$^2$R, especially in lengthy-context settings.

## 1 Introduction

Large language models (LLMs) have demonstrated remarkable capabilities across a wide spectrum of tasks, from question answering to complex reasoning and generation (Feng et al., 2023; Liu et al., 2024; Zhang et al., 2025; Jin et al., 2025). Despite such impressive progress, even the most capable LLMs, such as OpenAI-o1 (Achiam et al., 2023) and DeepSeek-R1 (Guo et al., 2025), still suffer from knowledge hallucination, i.e., producing factually incorrect yet seemingly plausible content. Recent advances in reasoning-oriented LLMs suggest that explicit reasoning processes can partially mitigate hallucination by enforcing more faithful intermediate steps. Nevertheless, in long-form tasks that require generating multiple sentences or paragraphs, hallucination tends to be further exacerbated (He et al., 2023; Xu et al., 2023; Wu et al., 2024; Cheng et al., 2025a).

To alleviate hallucination, retrieval-augmented language models (RALMs) have recently emerged as a promising paradigm (Vu et al., 2023; Yu et al., 2023). By incorporating external knowledge in a plug-and-play fashion, RALMs are able to complement the parametric memory of LLMs with accurate and up-to-date information. A growing body of work has demonstrated their effectiveness and this mechanism significantly reduces the reliance on potentially outdated or incomplete parametric knowledge, thereby mitigating hallucination (Gao et al., 2023; Wang et al., 2024).

However, RALMs are far from solving hallucination in long-form generation (Liu et al., 2025b; Chang et al., 2025b). A key challenge, which we refer to as **Lost in Lengthy Contexts**, arises when key evidence is obscured in long contexts. This challenge manifests in two aspects. First, retrieved results are often lengthy, and the redundant information makes it difficult for the model to capture

---

* Equal contribution.
† Corresponding author.

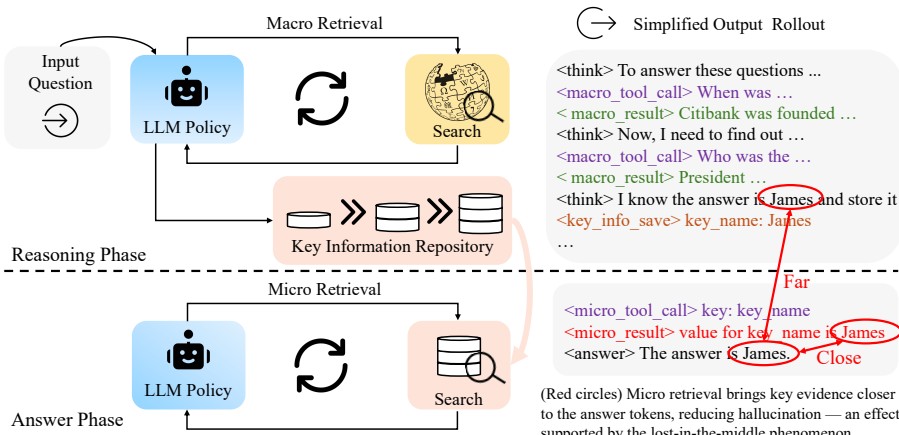

Figure 1: **Overview of the M²R framework.** During the reasoning phase, M²R performs macro retrieval and stores answer-aligned facts into an internal key-information repository. During the answer phase, the model invokes micro retrieval to fetch the stored facts and place them close to the generated answer tokens.

the key information (***Limitation 1***). Second, long reasoning chains often cause the model to forget earlier intermediate results, leading to errors in the final answer (***Limitation 2***).

Recent studies highlight that the *proximity* of key evidence to the final output is crucial for factual reliability: the closer the evidence appears to the final answer, the more likely the model is to remain faithful (Liu et al., 2023; Zhang et al., 2024). Additional empirical results and theoretical analysis of this phenomenon are provided in Appendix B. However, existing RALMs lack effective mechanisms to guarantee such proximity — external knowledge is injected into the reasoning process via multi-turn retrieval, but this strategy cannot ensure that essential evidence is retained near the outputs.

To overcome these limitations, we propose a **M**icro–**M**acro **R**etrieval (**M²R**) framework. As shown in Fig. 1, M²R has two components. The first is *macro retrieval*, which follows the traditional paradigm of retrieving relevant passages from external sources during the reasoning phase. Crucially, whenever the reasoning process yields answer-aligned evidence, it is preserved into a structured key–value repository, forming the *key information repository*, and the detection and storage of such key information are performed directly by the model during the `<think>` phase. The second is a novel *micro retrieval* mechanism introduced in the answer phase, which extracts essential results from this repository to ground the final output. By storing key information in a dedicated repository, the model avoids forgetting earlier intermediate results (***addressing Limitation 1***) while establishing a bridge that links macro retrieval with micro retrieval. During answer generation, the model can re-access the saved results and insert them directly before producing the corresponding output tokens. In this way, the proximity between key information and generated outputs is ensured, keeping key information tightly coupled with the answer (***addressing Limitation 2***). Finally, by adopting the retrieve-while-generate paradigm, M²R effectively alleviates hallucination in long-form tasks.

In terms of implementation, we employ a curriculum learning–based (Bengio et al., 2009) reinforcement learning (RL) strategy (i.e., GRPO (Shao et al., 2024)) to train the model to perform the entire micro–macro retrieval process. Customized rule-based rewards are designed to encourage accurate evidence saving and consistent grounding, allowing the model to gradually acquire the retrieval–reasoning skills in a stable manner. We train M²R from scratch on Qwen2.5-3B-Instruct (Hui et al., 2024) and Qwen2.5-7B-Instruct, and conduct extensive experiments on long-form question answering and retrieval-augmented generation benchmarks. Results demonstrate that M²R yields substantial improvements over strong baselines, with particularly pronounced gains under lengthy-context settings. Our contributions are summarized as follows:

- By grounding generation on position-aware key information, we propose the M²R framework. M²R introduces a new retrieve-while-generate mechanism during the answer phase, where re-

trieval is performed over model-generated key information, and answer generation is constrained by enforcing proximity between the retrieved evidence and the generated tokens.

- By employing a curriculum learning–based reinforcement learning strategy with customized rule-based rewards, M$^2$R gradually acquires the ability to progress from macro retrieval to key information saving and finally to micro retrieval in a stable manner.

- By conducting extensive experiments on different open-source benchmarks, M$^2$R demonstrates substantial improvements over strong baselines in terms of factual consistency and hallucination reduction, with particularly pronounced gains under lengthy-context settings.

## 2    RELATED WORK

LLMs have demonstrated outstanding performance across various tasks. However, in certain specialised domains or knowledge-intensive tasks, LLMs are prone to hallucinations. Regarding this problem, many approaches focus on detecting hallucinations in LLMs Wei et al. (2024); Kim et al. (2024b); Chuang et al. (2024); Luo et al. (2025); Zhong & Litman (2025). Recently, numerous methods for detecting hallucinations in LLMs have emerged, specifically targeting scenarios with long context (Feng et al., 2024; Shi et al., 2024a; Qin et al., 2025). Liu et al. (2025a) employed self-generated thoughts derived from preceding utterances as expressions to induce intrinsic knowledge and comprehend long-context semantics. Park et al. (2025) achieve hallucination detection by incorporating learnable lightweight and flexible steering vectors within LLMs.

Existing approaches to mitigating hallucinations in large language models can broadly be divided into two categories. One category comprises retrieval-augmented generation (RAG) (Izacard & Grave, 2021; Yu et al., 2024a; Xu et al., 2025; Izacard et al., 2023; Shi et al., 2024b; Li et al., 2024), which direct models to retrieve external knowledge, thereby enhancing response accuracy and reducing hallucinations. Numerous approaches have been developed to optimise the retrieval process for LLMs, thereby enhancing their performance. For instance, approaches such as Trivedi et al. (2023b), Shao et al. (2023b), and Yu et al. (2024b) introduce iterative retrieval-generation cycles, enabling LLMs to dynamically refine their retrieval strategies. Xu et al. (2024) and Kim et al. (2024a) enhance the utilisation of external information, reduce information overload, and improve factual consistency by optimising LLM generation through summarisation retrieval. Another class of approaches focuses on stimulating the LLM's capacity to utilise its internal knowledge. For instance, Li et al. (2023) and Chen et al. (2024) employ probes or learnable parameters to optimise feature representations within the LLM. Chang et al. (2025a) imposes constraints on the generative process of LLMs. Cheng et al. (2025b) implemented a slow-thinking generation process for LLMs through a tree-search-based algorithm, thereby reducing hallucinations during the reasoning process.

Prior multi-turn retrieval frameworks such as ReAct (Yao et al., 2023) and Self-RAG (Asai et al., 2023) interleave retrieval with generation, but they operate only over external documents and cannot access model-generated intermediate reasoning. In contrast, $M^2R$ retrieves from an internal key-information repository constructed during the reasoning phase, enabling reuse of model-generated evidence. Moreover, $M^2R$ explicitly enforces evidence proximity by placing retrieved key facts immediately before answer tokens, mitigating long-context drift, a constraint absent in prior methods.

## 3    METHOD

Our framework performs *macro retrieval* during reasoning to gather coarse evidence, and *micro retrieval* during answering to query a key-information repository at generation time. With GRPO-based RL training, the model learns to maintain crucial evidence close to the produced outputs, improving factual reliability in lengthy contexts.

In this section, we first introduce reinforcement learning with integrated micro–macro retrieval (§3.1). We then detail the micro–macro retrieval process itself, including the design of the training template and the rule-based reward modeling (§3.2 - §3.3). Finally, we present a curriculum learning-based training schedule that stabilizes M$^2$R training (§3.4).

## 3.1 REINFORCEMENT LEARNING WITH MICRO–MACRO RETRIEVAL

We formulate the RL objective under the proposed micro–macro retrieval framework as follows:

$$\max_{\pi_\theta} \; \mathbb{E}_{x\sim\mathcal{D}, y\sim\pi_\theta(\cdot|x;\mathcal{R}_{\mathrm{macro}},\mathcal{R}_{\mathrm{micro}})} \Big[ r_\phi(x,y) \Big] \;-\; \beta\,\mathbb{D}_{\mathrm{KL}}\Big[ \pi_\theta(y\mid x;\mathcal{R}_{\mathrm{macro}},\mathcal{R}_{\mathrm{micro}}) \\ \|\,\pi_{\mathrm{ref}}(y\mid x;\mathcal{R}_{\mathrm{macro}},\mathcal{R}_{\mathrm{micro}})\Big], \tag{1}$$

where $\pi_\theta$ is the policy LLM, $\pi_{\mathrm{ref}}$ is the reference LLM, $r_\phi$ is the rule-based reward function, and $\mathbb{D}_{\mathrm{KL}}$ is the KL-divergence regularizer. Here, $x$ denotes input samples from the dataset $\mathcal{D}$, and $y$ represents generated outputs conditioned on *macro retrieval* results $\mathcal{R}_{\mathrm{macro}}$ from external sources and *micro retrieval* results $\mathcal{R}_{\mathrm{micro}}$ from the key information repository constructed during reasoning.

Unlike prior retrieval-augmented RL approaches (Chen et al., 2025; Jin et al., 2025), our framework integrates two-level retrieval directly into the policy with a fixed *macro→micro* order:

$$\pi_\theta(\cdot\mid x;\mathcal{R}_{\mathrm{macro}},\mathcal{R}_{\mathrm{micro}}) = \pi_\theta^{\mathrm{answer}}(\cdot\mid x,\mathcal{M};\mathcal{R}_{\mathrm{micro}})\,\circ\,\pi_\theta^{\mathrm{think}}(\cdot\mid x;\mathcal{R}_{\mathrm{macro}}), \\ \mathcal{M} = \mathrm{SaveKey}\big(\pi_\theta^{\mathrm{think}}(\cdot\mid x;\mathcal{R}_{\mathrm{macro}})\big), \tag{2}$$

where $\circ$ denotes **sequential (staged) composition**: the policy first executes the `<think>` phase with *macro retrieval* to collect coarse-grained evidence and build the key-information repository $\mathcal{M}$, and then runs the `<answer>` phase with *micro retrieval* over $\mathcal{M}$. This staged policy leverages external evidence while keeping key information proximal to the final answer, leading to more reliable long-form generation.

**GRPO with Micro–Macro Retrieval.** We adopt *Group Relative Policy Optimization* (GRPO) as our RL algorithm. Unlike Proximal Policy Optimization (PPO), which typically trains an auxiliary value critic, GRPO estimates the baseline from a group of rollouts and therefore avoids an explicit critic. Given a current policy $\pi_{\theta_{\mathrm{old}}}$ and a fixed reference $\pi_{\theta_{\mathrm{ref}}}$, GRPO draws $G$ rollouts $\{y_i\}_{i=1}^G$ per input $x\sim\mathcal{D}$. The objective is:

$$\mathcal{J}(\theta) = \mathbb{E}_{x\sim\mathcal{D},\{y_i\}_{i=1}^G\sim\pi_{\theta_{\mathrm{old}}}(\cdot|x)}$$

$$\frac{1}{G}\sum_{i=1}^G \left[ \min\left( \frac{\pi_\theta(y_i|x)}{\pi_{\theta_{\mathrm{old}}}(y_i|x)} A_i, \mathrm{clip}\left( \frac{\pi_\theta(y_i|x)}{\pi_{\theta_{\mathrm{old}}}(y_i|x)}, 1-\epsilon, 1+\epsilon \right) A_i \right) - \beta\mathbb{D}_{\mathrm{KL}}\left(\pi_\theta||\pi_{\theta_{\mathrm{ref}}}\right) \right], \tag{3}$$

where $A_i = \big(r_i - \mathrm{mean}(\{r_j\}_{j=1}^G)\big)/\mathrm{std}(\{r_j\}_{j=1}^G)$ denotes the normalized advantage of the $i$-th rollout within the group, $\epsilon$ is the clipping threshold, and $\beta$ is the coefficient for the KL regularization term. The additional KL penalty prevents the updated policy from drifting too far from the reference model, stabilizing training and maintaining alignment with the base LLM. In our micro–macro retrieval framework, all policy terms in Eq. equation 3 are evaluated under retrieval conditioning; concretely, replace every occurrence of $\pi_\bullet(\cdot\mid x)$ with $\pi_\bullet(\cdot\mid x;\mathcal{R}_{\mathrm{macro}},\mathcal{R}_{\mathrm{micro}})$ (for $\bullet\in\{\theta,\theta_{\mathrm{old}},\theta_{\mathrm{ref}}\}$).

**Rollout with Macro and Micro Retrieval.** Unlike conventional rollouts that contain text-only reasoning, rollouts in M²R are *staged* as macro→micro. During the `<think>` phase, the policy may issue multiple macro retrieval calls (e.g., `<macro_tool_call>`) to external sources. Crucially, it saves *answer-aligned key information* (i.e., the answer to a specific question) into a structured key–value repository $\mathcal{M}$ using the `<key_info_save>` tag. In the subsequent `<answer>` phase, the policy performs micro retrieval calls (e.g., `<micro_tool_call>`) by querying $\mathcal{M}$ and conditions decoding on the returned values so that key information remains proximal to the output tokens and reduces hallucination in long-form generation.

**Retrieval Result Masking.** In standard GRPO, the policy loss is computed over all tokens in a rollout. In our setting, however, rollouts contain retrieval results that are injected by the environment (external tools) rather than produced by the policy. To avoid assigning credit to tokens the policy did not generate, we exclude retrieval-result spans when computing the loss. Concretely, in Eq. 3 we update gradients only on tokens corresponding to text-based reasoning and the model's own retrieval queries, while tokens inside retrieval results are masked out.

For implementation, let $m_t \in \{0, 1\}$ be a binary mask (1 for policy-generated tokens; 0 for retrieval results). We replace sequence log-prob terms with a masked sum,

$$\log \pi_\theta(y \mid \cdot) \stackrel{\triangle}{=} \sum_t m_t \log \pi_\theta(y_t \mid y_{<t}, \cdot) \big/ \max(1, \sum_t m_t), \tag{4}$$

and analogously form the (masked) log-ratio in Eq. 3. This preserves correct credit assignment, prevents spurious gradients from environment-injected text, and stabilizes training.

**What GRPO Optimizes in M²R.** It is important to clarify that GRPO in M²R does not optimize the retrieval module itself. Instead, GRPO supervises the model's generation behavior, teaching it (i) when to invoke macro- and micro-retrieval, (ii) how to compose and sequence tool calls, (iii) what key information should be written into the repository during the reasoning process, and (iv) how retrieved information should be incorporated into the final answer. Since the retrieval component is not modified by GRPO, M²R remains agnostic to the underlying retrieval system and is compatible with future improvements in retrieval quality.

## 3.2 TRAINING TEMPLATE

We describe the training template for both macro and micro retrieval within our framework. The training process is organized into two stages: macro retrieval in the `<think>` phase and micro retrieval in the `<answer>` phase. The complete prompt template is shown in Table 16.

**Macro Retrieval and Key Information Saving.** During the `<think>` phase, the model issues multi-turn macro retrieval calls enclosed within `<macro_tool_call>` tags. The purpose of these macro calls is to gather coarse-grained evidence from external sources. After retrieving the relevant information, the model saves the results as key-value pairs using the `<key_info_save>` tag, storing them in a structured repository $\mathcal{M}$, which is accessed later during the `<answer>` phase.

**Micro Retrieval for Final Answer Generation.** In the `<answer>` phase, the model queries $\mathcal{M}$ using the `<micro_tool_call>` tag. The retrieved results are returned within the `<micro_response>` tags, which are then used to form the final response. The final answer must be directly grounded on the results of micro retrieval. This ensures that the answer is based solely on the key information retrieved, rather than independent reasoning.

## 3.3 REWARD MODELING

Since there is no supervised reasoning data available, we design a rule-based reward function to optimize the policy through reinforcement learning. Our reward modeling consists of two primary components: *format reward* and *answer reward*.

- **Format Reward:** The format reward ensures that the model adheres to the predefined structure specified in the prompt templates for both macro and micro retrievals. Specifically, it checks the correctness of tag usage (e.g., valid `<macro_tool_call>` and `<key_info_save>` during reasoning, and `<micro_tool_call>` in the answer phase). It also ensures that every key value in the final answer is enclosed in `\boxed{}`.

- **Answer Reward:** The answer reward is a combination of three sub-rewards, all of which are computed using the F1 score:
  - *Final Answer Correctness ($s_{final}$):* This evaluates the agreement between the model's final output (values extracted from `\boxed{}`) and the ground-truth answer.
  - *Key Information Correctness ($s_{key}$):* This measures whether the key information stored in the key-value repository $\mathcal{M}$ aligns with the ground-truth answer, ensuring that only the most relevant evidence is retained.
  - *Consistency Score ($s_{cons}$):* This assesses the alignment between the stored key information and the final output, ensuring that the answer is grounded in the relevant retrieved evidence.

The total answer reward is computed as:

$$r_{\text{ans}} = s_{\text{final}} + \alpha\, s_{\text{key}} + \beta\, s_{\text{cons}}, \tag{5}$$

Specifically, for the final reward of a rollout:

$$r = \begin{cases} r_{\text{ans}}, & \text{if F1 score is not 0 and answer is correct,} \\ 0.1, & \text{if F1 score is 0 but format is correct,} \\ 0, & \text{if F1 score is 0 and format is incorrect.} \end{cases} \quad (6)$$

### 3.4 STABILIZING TRAINING WITH CURRICULUM LEARNING

Training a model to integrate macro retrieval, key information saving, and micro retrieval is challenging. In our initial experiments, we found that directly optimizing all components at once often leads to unstable rollouts and poor convergence. To address this issue, we employ a curriculum learning approach, dividing the training into two stages. In the first stage, the model focuses exclusively on macro retrieval and key information saving, learning to correctly identify relevant information and store it in the predefined structure. In the second stage, we introduce micro retrieval and fine-grained answer grounding, enabling the model to leverage the stored key information when generating the final response.

This staged training strategy offers several advantages. First, it simplifies the learning process by reducing the complexity at each stage, which leads to improved training stability. Second, it allows the model to progressively build the necessary skills, ensuring that the later micro retrieval steps are built upon a strong foundation of accurate macro retrieval and evidence saving. Finally, this approach encourages the model to generate answers that are not only factually accurate but also grounded in the retrieved evidence, maintaining consistency throughout the reasoning process.

This staged progression mirrors human reasoning, where individuals typically gather and organize broad information first, and then refine it into precise and reliable answers.

## 4 EXPERIMENT

To assess the effectiveness of M$^2$R, we conduct extensive experiments on multi-hop question answering benchmarks that demand multi-step reasoning and repeated retrieval (Feng et al., 2025b). These settings naturally induce *Lost in Lengthy Contexts* scenarios. Our method is instantiated on Qwen2.5-3B-Instruct and Qwen2.5-7B-Instruct. Following *ReSearch* (Chen et al., 2025), we train only on the MuSiQue (Trivedi et al., 2022) training split, which offers diverse multi-hop questions curated with fine-grained quality control. Our code is available at `https://github.com/WoodScene/M2R`.

**Benchmarks** We evaluate M$^2$R on four standard multi-hop QA benchmarks: HotpotQA (Yang et al., 2018), 2WikiMultiHopQA (Ho et al., 2020), MuSiQue (Trivedi et al., 2022), and Bamboogle (Press et al., 2023). HotpotQA, 2WikiMultiHopQA, and MuSiQue are automatically constructed from Wikipedia or Wikidata (Vrandecic & Krötzsch, 2014) with different multi-hop mining strategies and crowd-sourced validation, while Bamboogle is a manually curated set of challenging two-hop questions. For standard evaluation, we use the full development sets of HotpotQA (7,405), 2WikiMultiHopQA (12,576), MuSiQue (2,417), and the test set of Bamboogle (125). For the first three benchmarks, we discard the original contexts and only retain question–answer pairs, with retrieval performed from a shared Wikipedia corpus.

**Baselines** We compare M$^2$R against several baselines: (1) **No RAG**: directly using the instruction-tuned model to generate answers without retrieval augmentation; (2) **Naive RAG**: a standard retrieval-augmented setup where the retrieved documents are concatenated with the question before generation; (3) **Iter-RetGen** (Shao et al., 2023a): an iterative method that interleaves retrieval and generation; (4) **IRCoT** (Trivedi et al., 2023a): an iterleaving method, which use retrieval and the chain-of-thought (CoT) guide each other. (5) **COFT** (Lv et al., 2024): a coarse-to-fine framework that highlights key reference contexts to mitigate the problem of getting lost in lengthy inputs. (6) **SURE** (Kim et al., 2024b): generates summaries of retrieved passages for multiple answer candidates, and then selects the most plausible answer by evaluating and ranking these summaries. (7) **ReSearch** (Chen et al., 2025): a reinforcement learning–based framework that trains LLMs to reason with multi-turn search, serving as a strong baseline.

**Evaluation Metrics**  To assess the correctness of the final answers, we adopt two complementary metrics. First, we report Exact Match (*EM*), which considers a prediction correct only if it exactly matches the ground-truth answer. While straightforward, EM is often too rigid for our setting, since the retrieval environment is open-ended and the generated answers are expressed in natural language. To address this limitation, we further employ an LLM-as-a-judge (*LJ*) metric. Specifically, we use `gpt-4o-mini` with a tailored judging prompt to evaluate whether a predicted answer is semantically consistent with the ground truth. The full judge prompt is provided in Appendi C.

## 5 RESEARCH QUESTIONS

### RQ 1: ANSWER CORRECTNESS

*Does $M^2R$ improve answer correctness compared to existing RAG methods on multi-hop QA tasks?*

The main results of baselines and ReSearch are demonstrated in Table 1, and we show the methods based on LLMs with different sizes respectively. Compared with all baseline methods, $M^2R$ consistently achieves superior performance across multi-hop QA benchmarks, demonstrating the effectiveness of integrating both macro and micro retrieval. Specifically, $M^2R$ significantly outperforms the strongest baseline, *ReSearch*, which performs retrieval solely during the `<think>` phase.

These results verify that explicitly re-accessing key information through micro retrieval not only improves factual grounding but also enhances the correctness of the final output.

Table 1: Exact Match (EM, %) and LLM-as-a-Judge (LJ, %) results on multi-hop question answering benchmarks. The best results are highlighted in bold.

| Model | HotpotQA | | 2Wiki | | MuSiQue | | Bamboogle | |
|---|---|---|---|---|---|---|---|---|
| | EM | LJ | EM | LJ | EM | LJ | EM | LJ |
| **Qwen2.5-3B-Instruct** | | | | | | | | |
| Naive Generation | 12.05 | 18.45 | 10.09 | 19.79 | 2.62 | 6.08 | 5.01 | 8.50 |
| Naive RAG | 21.04 | 37.23 | 13.82 | 23.08 | 4.14 | 10.32 | 14.43 | 20.00 |
| Iter-RetGen | 24.63 | 42.22 | 14.75 | 28.86 | 6.91 | 13.43 | 16.03 | 22.61 |
| IRCoT | 23.63 | 40.60 | 12.50 | 23.54 | 4.39 | 10.83 | 17.60 | 26.13 |
| COFT | 36.17 | 50.88 | 32.82 | 39.76 | 13.49 | 20.11 | 25.30 | 31.38 |
| SURE | 35.44 | 51.23 | 36.20 | 42.38 | 17.24 | 27.61 | 31.20 | 39.95 |
| ReSearch | **38.78** | 55.70 | 38.90 | 47.41 | 19.40 | 31.56 | 38.11 | **48.12** |
| $M^2R$-Qwen-3B-Instruct | 38.70 | **56.46** | **40.07** | **48.34** | **20.87** | **32.97** | **39.58** | 47.20 |
| **Qwen2.5-7B-Instruct** | | | | | | | | |
| Naive Generation | 19.18 | 30.64 | 25.76 | 27.87 | 3.76 | 10.38 | 10.40 | 22.40 |
| Naive RAG | 31.90 | 49.59 | 25.78 | 29.52 | 6.21 | 12.78 | 20.80 | 32.00 |
| Iter-RetGen | 34.36 | 52.22 | 27.92 | 31.86 | 8.69 | 16.14 | 21.60 | 35.20 |
| IRCoT | 30.33 | 52.06 | 21.57 | 30.65 | 6.99 | 14.19 | 24.80 | 36.80 |
| COFT | 41.08 | 61.71 | 41.86 | 48.70 | 17.12 | 26.28 | 35.71 | 49.23 |
| SURE | 39.56 | 60.16 | 45.65 | 53.93 | 20.87 | 32.24 | 39.58 | 52.81 |
| ReSearch | 43.52 | 63.62 | 47.59 | 54.22 | 22.30 | 33.43 | 42.40 | 54.40 |
| $M^2R$-Qwen-7B-Instruct | **44.11** | **65.98** | **48.89** | **57.01** | **24.12** | **35.44** | **44.56** | **56.89** |

### RQ 2: HALLUCINATION REDUCTION

*Can $M^2R$ effectively reduce hallucinations under more challenging long-context scenarios?*

To further stress-test the ability of $M^2R$, we move beyond standard single-question inference and construct harder evaluation settings on the HotpotQA dataset. Specifically, we concatenate multiple questions into a single inference instance—denoted as HotpotQA-2Q (two questions) and HotpotQA-3Q (three questions)—requiring the model to answer them jointly within one rollout.

This setting substantially increases reasoning depth, retrieval calls, and contextual redundancy, thereby amplifying the difficulty of maintaining factual consistency.

We evaluate this setting with Qwen2.5-3B-Instruct, and results are shown in Figure 2. $M^2R$ consistently outperforms all baselines as the number of questions increases. While naive RAG and ReSearch suffer from rapidly rising hallucination rates, $M^2R$ maintains stable accuracy and substantially lower hallucinations. This robustness comes from its retrieve-while-generate paradigm, where micro retrieval continually re-anchors key evidence close to the outputs, making $M^2R$ especially effective in high-redundancy, long-context scenarios.

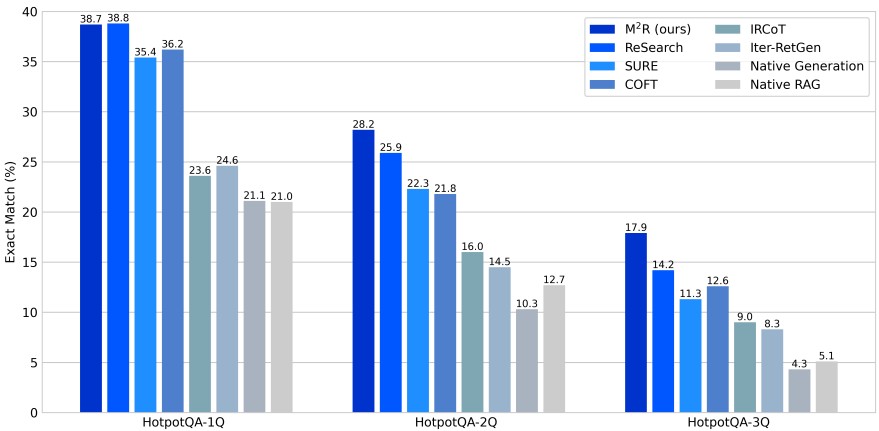

Figure 2: Performance on HotpotQA under multi-question inference settings. Here, HotpotQA-2Q denotes concatenating two questions into a single inference instance.

## RQ 3: ABLATION STUDY

*How critical is the retrieve-while-generate design in $M^2R$?*

To validate the contribution of micro retrieval, we compare our framework with a simplified variant that removes the retrieve-while-generate mechanism. In this baseline, when the model enters the `<answer>` phase, all saved key information from the repository $\mathcal{M}$ is provided to the model

Table 2: Ablation study on $M^2R$.

| Variant | EM (%) | LJ (%) |
|---|---|---|
| Full $M^2R$ | 24.12 | 35.44 |
| - One-shot Grounding | 23.38 | 34.72 |

at once. The model then generates the full answer based on this one-shot grounding, without invoking micro retrieval during generation. In contrast, $M^2R$ performs on-demand micro retrieval: at each step of answer generation, the model can selectively fetch only the relevant key information and re-anchor it immediately before producing the corresponding output tokens.

Results in Table 2 show that one-shot grounding yields weaker factual consistency, as injected evidence can be diluted by redundant reasoning tokens. In contrast, the retrieve-while-generate paradigm achieves more stable performance by inserting evidence precisely where needed. These results confirm that on-demand grounding is crucial for mitigating hallucination in long-form tasks.

## RQ 4: REWARD DYNAMICS

*How does $M^2R$ evolve in terms of reward during reinforcement learning?*

To further understand the training dynamics of $M^2R$, we analyze the reward curves during reinforcement learning, as shown in Figure 3. In the initial stage, the Qwen2.5-7B-Instruct model exhibits a much sharper increase in reward compared to the Qwen2.5-3B-Instruct model, demonstrating its stronger capacity to quickly adapt to the retrieve-while-generate paradigm. However, as training progresses, the reward growth of the Qwen2.5-3B-Instruct model gradually catches up, and both models eventually converge to a similar level. This suggests that while larger models can accelerate early adaptation, the long-term reward dynamics between different scales tend to align.

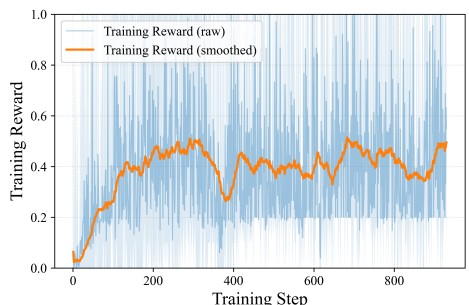 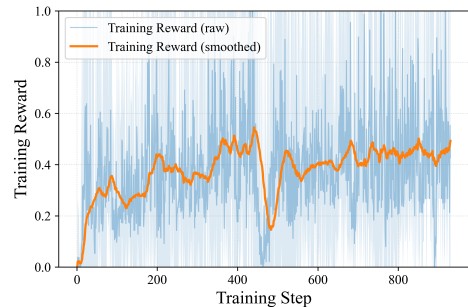

(a) Reward curve of M$^2$R with 3B backbone.       (b) Reward curve of M$^2$R with 7B backbone.

Figure 3: Reward dynamics of M$^2$R during reinforcement learning.

| Dataset | Think | Answer | Total | Min | Max |
|---|---|---|---|---|---|
| HotpotQA | 3.7 | 1.4 | 5.1 | 3 | 6 |
| 2Wiki | 4.5 | 1.7 | 6.2 | 3 | 10 |
| MuSiQue | 5.7 | 1.9 | 7.6 | 4 | 9 |
| Bamboogle | 3.5 | 1.3 | 4.8 | 2 | 6 |

Table 3: Average number of model invocations per query.

### RQ 5: CASE STUDY

To provide a clearer view of how M$^2$R operates in practice, Table 15 presents a simplified case study drawn from the evaluation set. This example demonstrates the reasoning and retrieval process of Qwen2.5-7B-Instruct under our framework. The text within `<think>` tags reflects the model's intermediate reasoning, while macro retrieval operations are invoked via `<macro_tool_call>` tags. Key evidence, directly aligned with the target answer, is explicitly preserved using `<key_info_save>` tags. In the final `<answer>` phase, the model invokes `<micro_tool_call>` to retrieve the stored key values, which are returned in `<micro_response>` and faithfully incorporated into the output. This case illustrates how M$^2$R decomposes a question into manageable steps, preserves essential evidence, and grounds the final prediction through micro retrieval. By positioning the supporting evidence close to the generated answer, the framework effectively reduces hallucinations and strengthens factual consistency.

### RQ 6: INFERENCE COST AND EFFICIENCY

*What is the inference-time overhead of M$^2$R, and how efficient is the micro–macro retrieval framework in practice?*

To understand the computational cost of M$^2$R, we measure (1) the number of model invocations per query, and (2) the end-to-end latency under standard inference settings. We separate the analysis into the `<think>` (macro retrieval) and `<answer>` (micro retrieval) phases.

**Model Invocations.** Table 3 reports the average number of model calls using Qwen2.5-3B-Instruct. Most invocations originate from the `<think>` phase—a cost shared by all multi-turn tool-based RAG frameworks. The additional overhead introduced by M$^2$R is only 1–2 micro-retrieval calls, corresponding to roughly a 20–30% relative increase. Micro retrieval itself is extremely lightweight, as it performs a rule-based lookup over a small, local repository.

**End-to-End Latency.** We benchmark real inference time, including all tool-calling and retrieval overhead, shown in Table 4. M$^2$R increases latency by less than 10% on average compared to ReSearch, while delivering significantly higher answer accuracy.

| Dataset | Avg Invocations | Inference Time (s) |
|---------|-----------------|--------------------|
| HotpotQA | 5.1 | $\approx$4.7 |
| 2Wiki | 6.2 | $\approx$5.2 |
| MuSiQue | 7.6 | $\approx$6.8 |
| Bamboogle | 4.8 | $\approx$4.6 |

Table 4: Measured inference time of $M^2R$ (Qwen2.5-3B + SGLang, 4×A100).

## 6 CONCLUSION AND FUTURE WORK

This work introduced Micro–Macro Retrieval ($M^2R$), a novel retrieve-while-generate framework that integrates *macro retrieval* during reasoning with *micro retrieval* during answering. By explicitly preserving and reusing key evidence close to the outputs, $M^2R$ directly addresses the "Lost in Lengthy Contexts" problem, leading to substantial gains in factual consistency and reduced hallucination over strong baselines. For future work, one direction is to move beyond simple rule-based rewards and incorporate learned reward models that better capture factuality, coherence, and grounding. Another is to further refine micro retrieval, for example by dynamically optimizing the proximity between evidence and output tokens. Finally, extending $M^2R$ with richer tool use, diverse external sources, and multimodal capabilities would broaden its applicability and robustness.

## ACKNOWLEDGMENTS

The authors thank the anonymous reviewers for their valuable feedback. This work was partially supported by Project P0056021 (Parent Project P0050643) of the Otto Poon Charitable Foundation Smart Cities Research Institute and by the Hong Kong General Research Fund (Grant No. 15204225).

## ETHICS STATEMENT

Our work focuses on improving the factual consistency and reliability of LLMs in multi-hop question answering and retrieval-augmented generation. All experiments are conducted on publicly available datasets (HotpotQA, 2WikiMultiHopQA, MuSiQue, and Bamboogle), which do not contain personally identifiable information or sensitive human-subject data. We do not introduce new data collection involving human participants. Our research complies with the ICLR Code of Ethics, and we see no direct risks regarding privacy, discrimination, or legal compliance.

## REPRODUCIBILITY STATEMENT

We make extensive efforts to ensure the reproducibility of our results.

- **Model and Training:** We describe the reinforcement learning setup (GRPO with micro–macro retrieval), training templates, and reward modeling details in Section 3, with additional hyperparameters in Appendix D.

- **Datasets:** All datasets used (HotpotQA, 2WikiMultiHopQA, MuSiQue, Bamboogle) are publicly available; we detail preprocessing and evaluation protocols in Section 4.

- **Code and Implementation:** To facilitate reproducibility, we provide an anonymous link to the source code and experimental scripts in the supplementary material.

Together, these measures allow independent researchers to reproduce our results and verify the claims made in this paper.

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

## A USE OF LARGE LANGUAGE MODELS (LLMS)

In preparing this work, we used large language models (LLMs) solely for improving the clarity and readability of the writing. Specifically, LLMs were employed as an assistant for language polishing, grammar checking, and style refinement. All research ideas, methodology design, experiments, analyses, and conclusions were conceived, implemented, and validated entirely by the authors without the involvement of LLMs. We take full responsibility for the content of this paper.

## B A POSITIONAL ENCODING PERSPECTIVE ON KEY-INFORMATION PROXIMITY

Empirical studies have shown that large language models often struggle to effectively use information located far from the prediction site, a phenomenon sometimes referred to as "lost in the middle" (Liu et al., 2023). As shown in Figure 4, the accuracy of GPT-3.5 on QA tasks decreases markedly when the answer-bearing document is positioned in the middle of the context. This observation underscores that the *proximity of key information to output tokens plays a critical role in ensuring factual reliability* (Dong et al., 2024; Zhou et al., 2026). Motivated by this observation, while the position of early inputs is largely fixed, we propose to actively adjust the placement of critical evidence closer to the output tokens, which directly inspired the design of our micro retrieval mechanism.

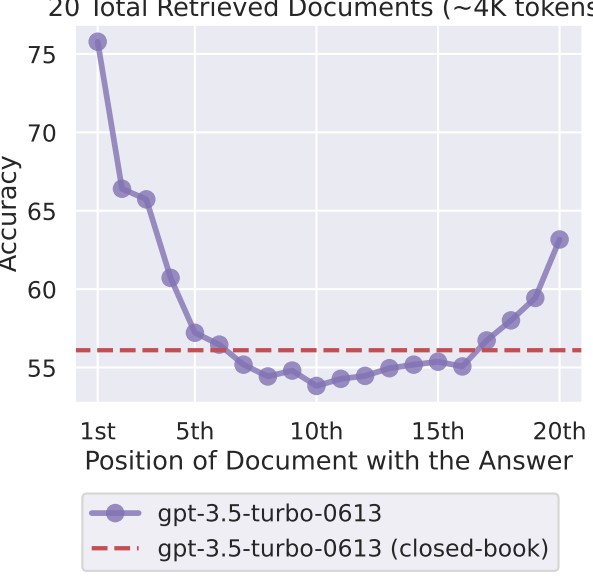

Figure 4: Effect of answer position on model accuracy (figure taken from Liu et al. (2023)). Accuracy declines sharply as the answer-bearing evidence appears in the middle of the context.

We also provide a theoretical explanation of this effect from a positional encoding perspective.

**RoPE and Relative Position Encoding.** Rotary Position Embeddings (RoPE) (Su et al., 2021) encode relative positions by rotating query and key vectors in the complex plane. For a query at position $m$ and key at position $n$, the inner product is:

$$q_m = R_{\theta,m} W_q x_m = R_{\theta,m} q, \quad k_n = R_{\theta,n} W_k x_n = R_{\theta,n} k, \tag{7}$$

$$q_m \cdot k_n = (R_{\theta,m} q)^\top (R_{\theta,n} k) = q^\top R_{\theta,m-n} k, \tag{8}$$

where $R_{\theta,m}$ is a block-diagonal rotation matrix with components

$$R_{\theta_i,m} = \begin{bmatrix} \cos(m\theta_i) & -\sin(m\theta_i) \\ \sin(m\theta_i) & \cos(m\theta_i) \end{bmatrix}, \quad \theta_i = b^{-\frac{2i}{d}}. \tag{9}$$

Table 5: Prompt for LLM-as-a-Judge.

| **Prompt Template** |
| --- |
| You will be given a question and its ground truth answer list where each item can be a ground truth answer. Provided a pred_answer, you need to judge if the pred_answer correctly answers the question based on the ground truth answer list. You should first give your rationale for the judgement, and then give your judgement result (i.e., correct or incorrect). 
 Here is the criteria for the judgement: 
 1. The pred_answer doesn't need to be exactly the same as any of the ground truth answers, but should be semantically same for the question. 
 2. Each item in the ground truth answer list can be viewed as a ground truth answer for the question, and the pred_answer should be semantically same to at least one of them. 
 question: {question} 
 ground truth answers: {gt_answer} 
 pred_answer: {pred_answer} 
 The output should in the following json format: 
 ```json 
 { 
     "rationale": "your rationale for the judgement, as a text", 
     "judgement": "your judgement result, can only be `correct` or `incorrect`" 
 } 
 ``` 
 Your output: |

**Spectral Decomposition.** The dot product can be decomposed into $d/2$ sinusoidal components with distinct frequencies $\theta_i$:

$$q_m \cdot k_n = \sum_{i=0}^{d/2-1} \big( q_{2i} k_{2i} \cos(\Delta \theta_i) + q_{2i+1} k_{2i+1} \sin(\Delta \theta_i) \big), \tag{10}$$

where $\Delta = m - n$ is the relative distance. High-frequency components oscillate rapidly and cancel out when $\Delta$ is large, while low-frequency components dominate at shorter distances. This spectral bias makes attention contributions stronger for nearby tokens than for distant ones.

**Proposition B.1.** *For RoPE-based attention, the expected contribution of evidence tokens decreases monotonically with their distance to the output position. Hence, key information placed closer to the outputs is more likely to be faithfully incorporated into generation, providing a theoretical justification for the effectiveness of key-information proximity.*

**Discussion.** This analysis shows that proximity is not only an empirical observation but also a theoretical consequence of how positional encoding interacts with attention. Placing key information closer to output tokens mitigates the risk of dilution by redundant context and reduces the chance of being forgotten in long reasoning chains. This provides a formal foundation for the design of our micro–macro retrieval framework, which explicitly manages evidence placement to improve factual consistency.

## C    PROMPT FOR LLM-AS-A-JUDGE

Table 5 presents the exact prompt we used to evaluate model responses under the LLM-as-a-Judge setting, ensuring consistency and reproducibility of the evaluation process.

## D    IMPLEMENTATION DETAILS

**Implementation Details.** We build our reinforcement learning framework upon `verl` (Sheng et al., 2024; Zhou et al., 2025). For training, we use the MuSiQue dataset, restricting to the training

split (19,938 samples), and train the models for two epochs. The retrieval environment is implemented with FlashRAG (Jin et al., 2024; Hui et al., 2025), a standard toolkit for retrieval-augmented generation. Following ReSearch, we adopt E5-base-v2 (Wang et al., 2022) as the dense retriever and use the December 2018 Wikipedia snapshot as the underlying knowledge base (Karpukhin et al., 2020; Ding et al., 2024; 2025). All document embeddings and indexes are preprocessed by FlashRAG (Luo et al., 2024; Hui et al., 2026). During both training and evaluation rollouts, we retrieve the top-5 passages for each query. For baseline systems, we directly use the implementations provided by FlashRAG to ensure fairness. In Eq. 5, we set $\alpha = \frac{1}{3}$ and $\beta = \frac{1}{10}$, as these values were found to provide a good balance between final answer correctness, key information preservation, and consistency after empirical validation in preliminary experiments (Kang et al., 2025).

To further improve reproducibility and transparency, we provide additional implementation details regarding hardware, environment configuration, and experimental settings. These specifications will also be included in the publicly released code.

**Hardware Requirements.** All models were trained on $8 \times$A100 40GB GPUs. All inference experiments were conducted on $4 \times$A100 40GB GPUs using the SGLang serving framework (Wu et al., 2025).

**Random Seeds.** All experiments were run with a fixed random seed of 42 to ensure determinism and reproducibility where possible.

These details ensure that future researchers can reliably reproduce both the training and inference pipelines of M$^2$R. We show some important parameter settings during training in Table 6.

Table 6: Implementation details of $M^2R$.

| Parameter | Value |
|---|---|
| Learning Rate | 1e-6 |
| Train Batch Size | 256 |
| Number of Training Epochs | 2 |
| Number of Rollout | 5 |
| Rollout Temperature | 1.0 |
| KL Loss Coefficient | 0.001 |
| Clip Ratio | 0.2 |

# E ADDITIONAL EXPERIMENTS

## E.1 EXTENDED MODEL FAMILIES AND MULTI-QUESTION REASONING BENCHMARKS

To further assess the empirical coverage of M$^2$R, we conduct additional experiments on (1) larger and different model families, and (2) more challenging long-form settings. These results complement the main experiments and address concerns regarding generalizability and training sufficiency.

**Different Model Families.** We evaluate M$^2$R on two additional models, Llama-3.1-8B-Instruct and Mistral-7B-Instruct. As shown in Table 7, M$^2$R consistently outperforms ReSearch across both model families, with an average improvement of 1.03%.

**Long-Form Multi-Question Benchmarks.** To evaluate the effectiveness of M$^2$R under longer reasoning chains, we extend each dataset by concatenating multiple questions (3Q and 5Q). As shown in Table 8, M$^2$R achieves the largest gains under these extended settings, validating the benefit of micro retrieval in maintaining localized evidence for deeper reasoning.

These additional results demonstrate that M$^2$R generalizes robustly across model families and remains effective under substantially longer reasoning chains.

| | Llama-3.1-8B-Instruct | | | |
|---|---|---|---|---|
| Method | HotpotQA | 2Wiki | MuSiQue | Bamboogle |
| Naive Generation | 18.1 | 24.7 | 4.4 | 11.6 |
| Naive RAG | 34.2 | 31.3 | 9.8 | 20.9 |
| COFT | 38.4 | 39.5 | 15.8 | 32.5 |
| SURE | 39.3 | 42.0 | 18.7 | 37.8 |
| ReSearch | 42.2 | 45.8 | 20.9 | **43.1** |
| **M²R** | **43.0** | **47.2** | **22.1** | 42.9 |
| | Mistral-7B-Instruct | | | |
| Method | HotpotQA | 2Wiki | MuSiQue | Bamboogle |
| Naive Generation | 21.9 | 28.8 | 7.7 | 12.5 |
| Naive RAG | 34.5 | 31.2 | 11.3 | 25.4 |
| COFT | 43.5 | 45.5 | 18.5 | 40.3 |
| SURE | 42.6 | 47.7 | 21.2 | 42.8 |
| ReSearch | 45.0 | 49.1 | 23.7 | 45.5 |
| **M²R** | **45.6** | **50.0** | **25.5** | **46.0** |

Table 7: Exact Match results on additional model families.

| Method | HotpotQA-3Q | 2Wiki-3Q | MuSiQue-3Q | Bamboogle-3Q |
|---|---|---|---|---|
| Naive Generation | 13.1 | 17.5 | 4.3 | 7.1 |
| Naive RAG | 20.6 | 16.9 | 5.1 | 15.2 |
| COFT | 24.5 | 27.2 | 12.6 | 22.0 |
| SURE | 28.3 | 31.5 | 11.3 | 25.5 |
| ReSearch | 30.2 | 33.9 | 14.2 | 28.0 |
| **M²R** | **32.0** | **35.8** | **17.9** | **30.6** |
| Method | HotpotQA-5Q | 2Wiki-5Q | MuSiQue-5Q | Bamboogle-5Q |
| Naive Generation | 5.5 | 4.5 | 0.7 | 1.8 |
| Naive RAG | 8.2 | 7.7 | 2.3 | 4.5 |
| COFT | 9.5 | 11.1 | 3.5 | 9.5 |
| SURE | 13.1 | 14.8 | 4.8 | 10.7 |
| ReSearch | 13.9 | 17.0 | 5.7 | 12.8 |
| **M²R** | **15.4** | **18.5** | **8.4** | **14.9** |

Table 8: Performance under multi-question reasoning (3Q and 5Q).

## E.2    ADDITIONAL ANALYSIS OF FLASHRAG CONFIGURATION AND RETRIEVAL ABLATIONS

This section provides additional details of the FlashRAG configuration, ablations on retrieval hyper-parameters, and token statistics during inference. These analyses complement the main results and demonstrate that M²R is robust to retrieval settings.

**FlashRAG Configuration.**    To ensure fair comparison and avoid introducing retrieval-side advantages, we strictly follow the official FlashRAG and Re-Search configuration without modification:

- **Knowledge Base:** FlashRAG's December 2018 Wikipedia snapshot.

- **Chunk Size:** ∼100-word passages (default).

- **Retriever:** E5-base-v2 dense retriever (Feng et al., 2025a).

This ensures that improvements from M²R stem from its generation-side retrieval mechanism rather than retrieval tuning.

**Ablations on Retrieval Settings.**    We ablate two key FlashRAG parameters—retrieve-top-$k$ and chunk size—on 2Wiki using Qwen2.5-3B. As shown in Table 9, M²R consistently outperforms ReSearch across all configurations, demonstrating strong robustness to retrieval hyperparameters.

| Retrieve-Top-$k$ | Naive RAG | ReSearch | $\mathbf{M^2R}$ |
|---|---|---|---|
| 3 | 13.5 | 37.2 | **38.3** |
| 5 (default) | 13.8 | 38.9 | **40.1** |
| 8 | 13.6 | 38.0 | **39.4** |

| Chunk Size | Naive RAG | ReSearch | $\mathbf{M^2R}$ |
|---|---|---|---|
| 50 | 13.4 | 38.1 | **39.4** |
| 100 (default) | 13.8 | 38.9 | **40.1** |
| 150 | 13.9 | 38.4 | **39.7** |

Table 9: Ablations on retrieve-top-$k$ and chunk size for FlashRAG.

| Dataset | Input Tokens | Output Tokens (ReSearch) | Output Tokens ($M^2R$) |
|---|---|---|---|
| HotpotQA | 25 | 416 | 432 |
| MuSiQue | 31 | 483 | 505 |
| 2Wiki | 37 | 440 | 478 |
| Bamboogle | 21 | 376 | 389 |

Table 10: Token statistics during inference. $M^2R$ produces slightly longer outputs due to micro retrieval, but the inserted key facts are compact and answer-aligned, improving grounding and final accuracy.

**Token Statistics During Inference.** **To analyze whether $\mathbf{M^2R}$ alleviates long-form reasoning constraints, we report input and output token statistics in Table 10. "Input Tokens" represent question tokens only, whereas "Output Tokens" include both reasoning chains and final answers (excluding retrieved passages).** Overall, these analyses indicate that $M^2R$ is robust to retrieval configurations and benefits long-form reasoning by injecting concise, model-generated key information near the answer generation step.

### E.3 Inference Cost and Storage Analysis

We provide additional analysis of the inference latency and storage cost of the key-information repository in $M^2R$. These results complement the main experiments and demonstrate that the micro–macro retrieval pipeline introduces only minimal overhead.

**Inference Latency.** The macro-retrieval stage in $M^2R$ follows the same workflow as standard RAG (Zhan et al., 2025; Feng et al., 2026), and the key-information saving step stores only a handful of answer-aligned facts (typically 3–10 items), making its cost negligible. Micro retrieval is also lightweight, as it performs a simple dictionary-style lookup over a small local repository.

To quantify the overhead, we report real inference time using Qwen2.5-3B with SGLang on $4 \times$ A100 40GB GPUs. As shown in Table 11, $M^2R$ increases inference time by less than 10% on average compared to ReSearch, while offering substantially larger accuracy improvements.

| Inference Time (s) | HotpotQA | 2Wiki | MuSiQue | Bamboogle |
|---|---|---|---|---|
| ReSearch | $\approx$4.3 | $\approx$4.8 | $\approx$6.3 | $\approx$4.2 |
| **$\mathbf{M^2R}$** | $\approx$**4.7** | $\approx$**5.2** | $\approx$**6.8** | $\approx$**4.6** |

Table 11: Measured inference latency of $M^2R$ compared to ReSearch.

**Scaling With Input Complexity.** **We also evaluate a multi-question setting by concatenating 2–3 HotpotQA questions into a single input (Table 12). Latency grows approximately linearly with reasoning complexity, consistent with multi-turn tool-use systems.** Overall, these results show that $M^2R$ introduces only minimal overhead beyond standard RAG systems. The micro–macro retrieval pipeline remains efficient even under long-form reasoning.

| Setting | Avg Invocations | Inference Time (s) |
|---------|-----------------|--------------------|
| 1Q | 7.6 | ≈6.8 |
| 2Q | 13.8 | ≈14.1 |
| 3Q | 19.7 | ≈22.3 |

Table 12: Latency scaling under multi-question reasoning.

**Storage Cost of the Key-Information Repository.** The key-information repository stores a very small number of atomic facts produced during the `<think>` phase. Table 13 reports the measured token counts. Across all datasets, the repository remains small (50–150 tokens), which is negligible compared with the retrieved passages themselves.

| Dataset | Avg Tokens | Min | Max |
|---------|-----------|-----|-----|
| HotpotQA | 62 | 24 | 108 |
| 2Wiki | 73 | 28 | 121 |
| MuSiQue | 88 | 32 | 139 |
| Bamboogle | 55 | 18 | 95 |

Table 13: Size of the key-information repository measured in tokens.

Overall, these results show that $M^2R$ introduces minimal inference overhead and negligible storage cost, while providing substantial improvements in grounding and answer correctness.

### E.4 Ablation Study: Importance of Curriculum Learning

To further analyze the role of curriculum learning in $M^2R$, we compare our two-stage training strategy with direct joint optimization of macro and micro retrieval. This experiment complements the main results and provides insight into training stability and optimization difficulty.

**Accuracy Comparison.** Table 14 reports the Exact Match scores under three training strategies. Direct optimization performs poorly across all datasets—even worse than Naive RAG—because the model must simultaneously learn macro retrieval, key-information saving, and micro retrieval. This substantially increases optimization difficulty and prevents the model from obtaining meaningful rewards.

| Training Strategy | HotpotQA | 2Wiki | MuSiQue | Bamboogle |
|-------------------|----------|-------|---------|-----------|
| Naive RAG | 21.0 | 13.8 | 4.1 | 14.3 |
| Direct Optimization | 13.2 | 8.4 | 3.9 | 10.8 |
| **Curriculum Learning (ours)** | **38.7** | **40.1** | **20.9** | **39.6** |

Table 14: Comparison of training strategies, showing the importance of curriculum learning.

**Discussion.** In contrast to direct optimization, curriculum learning decomposes training into two tractable stages, allowing the model to first master macro retrieval before learning micro retrieval. This staged formulation dramatically stabilizes training, reduces reward sparsity, and yields significantly better end-to-end performance. These findings validate the necessity of curriculum learning in effectively training $M^2R$.

### F Case Study

Table 15 presents a simplified case study drawn from the evaluation set.

### G Prompt Template for $M^2R$

The complete prompt template is shown in Table 16.

Table 15: A simplified case study of M$^2$R on a test instance.

---

**Question:** Guest 101 wants to book any available suite room and needs to know both the total price after VIP discount and the room number with the lowest payable amount.

**Answer:** 180.0, Room 301

---

`<think>` First, I need to find an available suite room. Then I will calculate the price and apply the VIP discount for Guest 101. `</think>`

`<macro_tool_call>`
  "name": "get_available_rooms", "room_type": "suite"
`</macro_tool_call>`

`<macro_result>` "number": 301, "type": "suite", "price": 200.0, "available": True; "number": 302, "type": "suite", "price": 220.0, "available": True `</macro_result>`

`<think>` Rooms 301 and 302 are available. Their prices are 200.0 and 220.0, respectively. With a 10% VIP discount, the payable amounts would be 180.0 and 198.0. `</think>`

`<macro_tool_call>`
  "name": "get_guest_vip_status", "guest_id": 101
`</macro_tool_call>`

`<macro_result>` True `</macro_result>`

`<think>` Guest 101 is confirmed as VIP. The lowest final payable amount is 180.0 for Room 301. `</think>`

`<key_info_save>`
  "finalPayableAmount": "180.0", "RoomNumber": "301"
`</key_info_save>`

`<think>` I have obtained both the final payable amount and the room number with the lowest cost. I will now proceed to the answering phase. `</think>`

`<answer>`
  `<micro_tool_call>`        "query":              "finalPayableAmount" `</micro_tool_call>`
  `<micro_response>`"finalPayableAmount":"180.0"`</micro_response>`
  The requested final payable amount is \boxed{180.0}.
  `<micro_tool_call>` "query": "RoomNumber" `</micro_tool_call>`
  `<micro_response>`"RoomNumber":"301"`</micro_response>`
  The suite room with the lowest payable amount is \boxed{301}.
`</answer>`

Table 16: Template for M$^2$R.

| Prompt Template For M$^2$R |
| --- |

You are a helpful assistant. The assistant first thinks about the reasoning process in the mind and then provides the user with the answer.

During the reasoning process, you have access to a set of tools you can use to assist with the user query, referred to as **macro retrievals**. These macro retrievals are enclosed within <macro_tool_call></macro_tool_call> tags. You may conduct multiple rounds of function calls, and in each round, you can call one or more functions.

The results of the macro function calls will be given back to you after execution, and you can continue to call functions until you get the final answer for the user's question.

Additionally, during the reasoning process, whenever you obtain the answer to the user's question, you **must store it as a key-value pair** in a key information dictionary using the <key_info_save></key_info_save> tag.

- The format must strictly follow JSON, e.g.: {"target_value": "value"}
- The stored key-value pairs must be directly relevant to the final answer.

Finally, once you have obtained the answer and stored the key information, proceed to the answering phase. At this stage, do not call any further functions. Before writing the final answer sentence, you must first perform **micro retrieval** to fetch the answer from the key information dictionary. The final answer must be based only on the results of micro retrieval, rather than answering independently.

Notes for micro retrieval:

- Micro retrieval must be enclosed within <micro_tool_call></micro_tool_call> tags.
- You may query multiple items at once or issue requests in batches.
- The results of micro retrieval will be provided after execution.
- If the micro retrieval fails, you must simply state that the result could not be retrieved, and must not fabricate an answer independently.

Every value from the key information dictionary that appears in the final answer must be enclosed in \boxed{}.

