# OpenReview forum: "Micro-Macro Retrieval: Reducing Long-Form Hallucination in Large Language Models"
_ICLR.cc/2026/Conference — ICLR 2026 Poster_

### Official Review · Reviewer_vzvj · 2025-10-25

**Soundness:** 3
**Presentation:** 4
**Contribution:** 3
**Rating:** 6
**Confidence:** 3

**Summary:**

This paper aims to reduce LLM hallucinations in long-context settings, especially in the context of RAG.
The key limitation of current RAG approaches is that, either the long context makes it hard for the model to accurately identify key info, or model fails to retrieve intermediate results from its own reasoning chains.
To address this limitation, the paper introduces **micro-macro retrieval (M$^2$R)**, a retrieve-while-generate method.
M$^2$R hinges on the positive correlation between key information proximity to model outputs and factual accuracy, which was reported previous works.
M$^2$R directly enforces this proximity mechanism into the LLM through curriculum-learning with GRPO, such that *macro retrieval* (also `<think>` phase) implements traditional RAG and maintains a key info repo while *micro retrieval* (also `<answer>` phase) extracts key info from the established repo to ground model outputs.

Experiment evaluation on multi-hop QA benchmarks show that M$^2$R generally outperforms most baselines and its comparative advantage is more evident in challenging scenarios (HotpotQA-2/3Q).

**Strengths:**

1. Originality

   This paper is novel in that it alleviates hallucinations in RAG based directly on empirical insights from [1-2] that key information position methods in long-form generation.
   The application of GRPO + curriculum learning to realize the insight above also makes sense.
2. Quality

   The experiment design is sound and directly supports central claims of this paper.
   The authors have also done an excellent job by releasing the source code with clear documentation.
3. Clarity

   The paper clearly explains its motivation, key insights (lines 53-55, Appendix B), as well as discussions on the rationale of key designs (lines 120-122, 136-137, 161, 249-256, etc).
   The case study is useful for readers to understand the method intuitively.
4. Significance

   This paper could be of significance to the field by grounding in empirical findings of previous works, combining GRPO and curriculum learning and achieve a compelling performance boost in challenging multi-hop QA tasks.

References:

[1] Lost in the middle: How language models use long contexts. (2023)
[2] Found in the middle: How language models use long contexts better via plug-and-play positional encoding. (2024)

**Weaknesses:**

(Authors do **not** need to refer to points raised in this section since the main points are already mentioned in *"Questions" section*.)
1. W1: Limited model family

   The paper only involves experiments on Qwen-2.5-3B/7B models; results on more diverse model families could strengthen the paper's arguments.
2. W2: Missing discussions on costs

   Since the core method introduces additional storage requirements (key info repo) and requires retrieving key info during answer phase, there are concerns regarding whether these components induce heavy storage/time costs.

**Questions:**

**Major questions (that could affect rating)**
1. **Question 1**: Limited model family

   M$^2$R is only tested on two models of different sizes (3B, 7B) but the same model family (Qwen2.5). This limitation does *not* directly undermine the central claims of the paper, but results on more diverse model families could greatly enhance the general utility of the proposed method.
2. **Question 2**: Cost analysis

   Inference efficiency is a critical concern in RAG applications. The proposed method requires maintaining a key info repo during macro retrieval and retrieving key infos during micro retrieval. Therefore, a natural concern arises as to whether the performance boost is worth the cost:

   *How does this micro-macro framework affect inference latency, and what are the storage costs of the key info repo?*

   A detailed analysis (either theoretical or empirical) could be useful in deciding whether M$^2$R is usable in practice.

**Minor questions and suggestions (that are not considered to affect rating)**
1. **Minor question 1**: Additional details for reproducibility

   Although the authors have provided source code and some experiment details in the paper, additional details such as hardware requirements and seeds could be useful for reproductions of results.
2. **Suggestion 1**: Notations

   The method name, M$^2$R, is not consistently presented: it is usually written in normal font but sometimes written in italics (lines 206-261, 267, etc).
3. **Suggestion 2**: Paper organization

   Related Work section could help readers set the context, but currently it is placed in the Appendix. Therefore I recommend move Table 1 (M$^2$R prompt template) to the Appendix and move Related Work section to the main body instead.
4. **Suggestion 3**: Details on training stability

   The paper mentions at lines 242-243 that, directly optimizing macro/micro retrieval leads to poor convergence. Detailed results (preferably placed in the Appendix) could help future researchers gain in-depth understandings regarding the significance of curriculum learning.

---

> ### Author Response · Authors · 2025-11-23
> **Response to Reviewer vzvj (Part 1)**
>
> Thank you for your positive and constructive feedback. We hope our responses below address your concerns and further increase your confidence in our work.
>
> ### Q1: Limited Model Family
> Thank you for the suggestion. To assess the generalizability of $M^2R$, we extend our evaluation to two additional model families: Llama-3.1-8B-Instruct and Mistral-7B-Instruct. Exact Match results are reported below.
>
> **Results on Llama-3.1-8B-Instruct**
> | Method           | HotpotQA | 2Wiki    | MuSiQue  | Bamboogle |
> | ---------------- | -------- | -------- | -------- | --------- |
> | Naive Generation | 18.1     | 24.7     | 4.4      | 11.6       |
> | Naive RAG        | 34.2     | 31.3     | 9.8      | 20.9      |
> | COFT             | 38.4     | 39.5     | 15.8     | 32.5      |
> | SURE             | 39.3     | 42.0     | 18.7     | 37.8      |
> | ReSearch         | 42.2     | 45.8     | 20.9     | **43.1**      |
> | **$M^2R$ (ours)**   | **43.0** | **47.2** | **22.1** | 42.9  |
>
>
> **Results on Mistral-7B-Instruct**
> | Method           | HotpotQA | 2Wiki    | MuSiQue  | Bamboogle |
> | ---------------- | -------- | -------- | -------- | --------- |
> | Naive Generation | 21.9     | 28.8     | 7.7      | 12.5      |
> | Naive RAG        | 34.5     | 31.2     | 11.3      | 25.4      |
> | COFT             | 43.5     | 45.5     | 18.5     | 40.3      |
> | SURE             | 42.6     | 47.7     | 21.2     | 42.8      |
> | ReSearch         | 45.0     | 49.1     | 23.7     | 45.5      |
> | **$M^2R$ (ours)**   | **45.6** | **50.0** | **25.5** | **46.0**  |
>
>
> Across both model families, $M^2R$ outperforms ReSearch in 7 out of 8 cases, highlighting its strong cross-model robustness.
>
>
> **Experiments on More Challenging Reasoning Benchmarks**
>
> To assess longer reasoning chains and more complex tool-use dynamics, we extend the evaluation by requiring the model to answer multiple questions in a single input (3Q or 5Q variants). Results on Qwen2.5-7B-Instruct are shown below.
>
>
> | Method           | HotpotQA-3Q | 2Wiki-3Q | MuSiQue-3Q | Bamboogle-3Q |
> | ---------------- | ----------- | -------- | ---------- | ------------ |
> | Naive Generation | 13.1         | 17.5      | 4.3        | 7.1          |
> | Naive RAG        | 20.6        | 16.9     | 5.1        | 15.2          |
> | COFT             | 24.5        | 27.2     | 12.6        | 22.0         |
> | SURE             | 28.3        | 31.5     | 11.3        | 25.5         |
> | ReSearch         | 30.2        | 33.9     | 14.2        | 28.0         |
> | **$M^2R$ (ours)**   | **32.0**    | **35.8** | **17.9**   | **30.6**     |
>
>
>
>
> | Method           | HotpotQA-5Q | 2Wiki-5Q | MuSiQue-5Q | Bamboogle-5Q |
> | ---------------- | ----------- | -------- | ---------- | ------------ |
> | Naive Generation | 5.5         | 4.5      | 0.7        | 1.8          |
> | Naive RAG        | 8.2         | 7.7      | 2.3        | 4.5          |
> | COFT             | 9.5         | 11.1     | 3.5        | 9.5          |
> | SURE             | 13.1        | 14.8     | 4.8        | 10.7         |
> | ReSearch         | 13.9        | 17.0     | 5.7        | 12.8         |
> | **$M^2R$ (ours)**   | **15.4**    | **18.5** | **8.4**    | **14.9**     |
>
>
> The results show that $M^2R$ achieves significant improvements under extended reasoning chains, effectively addressing the lost-in-lengthy-contexts issue and demonstrating strong performance in long-form reasoning tasks.
>
> We have included these additional experimental results in the revised manuscript, highlighted in blue (Lines 902-958).
>
>
> ### Q2: Cost Analysis
> > How does this micro-macro framework affect inference latency, and what are the storage costs of the key info repo?
>
> Thank you for the question. We report both inference latency and storage cost below.
>
> **(1) Inference Latency**
>
> The macro-retrieval stage in $M^2R$ follows the same workflow as standard RAG, so it introduces no additional time complexity.
> The key-information saving step stores only a handful of answer-aligned facts (typically 2–10 items), so its cost is negligible.
>
> **Micro retrieval is also lightweight**, as it performs a simple dictionary-style lookup over this local small repository.
>
> To quantify the overhead, we report actual inference time using Qwen2.5-3B + SGLang on 4×A100 GPUs:
>
> | Inference Time Cost   |  HotpotQA  |  2Wiki  | MuSiQue | Bamboogle |
> |---------------------|------|------|------|-----|
> | ReSearch |$\approx$  4.3 s | $\approx$  4.8 s | $\approx$  6.3 s | $\approx$  4.2 s
> | $M^2R$ (ours)  |   **$\approx$  4.7 s**                |   **$\approx$  5.2 s**              | **$\approx$  6.8 s**             |  **$\approx$  4.6 s**
>
> $M^2R$ increases inference time by less than 10% on average compared to ReSearch, while providing substantially larger accuracy improvements. This confirms that micro-level grounding adds only minimal latency.

---

> ### Author Response · Authors · 2025-11-23
> **Response to Reviewer vzvj (Part 2)**
>
> **(2) Storage Cost of the Key-Information Repository**
>
> The key-information repository stores a very small number of atomic facts produced during the \<think\> phase. We report the measured token count below:
>
> | **Key-Info Repository Size** | **Avg #Tokens** | **Min** | **Max** |
> | ---------------------------- | --------------- | ------- | ------- |
> | **HotpotQA**                 | **62**          | 24      | 108     |
> | **2Wiki**                    | **73**          | 28      | 121     |
> | **MuSiQue**                  | **88**          | 32      | 139     |
> | **Bamboogle**                | **55**          | 18      | 95      |
>
> Across all datasets, the repository size remains extremely small (~50–150 tokens), which is negligible compared to the retrieved context itself. Thus, storage cost is minimal.
>
> We have incorporated this latency and storage analysis into the revised manuscript (Lines 512–522, Lines 1054-1068).
>
>
> ### Q3: Additional details for reproducibility
> Thank you for your suggestion. In the revised manuscript (Lines 854–863), we have included the following items:
>
> * **Hardware requirements:**
>   * Training conducted on 8×A100 40GB GPUs.
>   * Inference performed on 4×A100 40GB GPUs using SGLang.
> * **Random seeds:** all experiments used a fixed seed of 42.
> * **Environment configuration:** Python version, dependency versions, and all required libraries are fully specified in the released requirements.txt.
>
> These additions ensure that future researchers can reliably reproduce both our training and inference pipelines.
>
>
>
> ### **Response to Suggestions**
> Thank you for the valuable suggestions. We have corrected the notation inconsistencies of the method name and reorganized the manuscript accordingly: the Related Work section has been moved into the main body, and Table 1 (the prompt template) has been placed in the Appendix.
>
> We also include additional results to illustrate the importance of curriculum learning (Lines 1072–1128). Specifically, we compare direct joint optimization of macro/micro retrieval with our curriculum learning strategy, analyzing both the training reward trajectories and the final performance. The accuracy comparison is shown below:
>
> | Training Strategy          |   HotpotQA  |  2Wiki  | MuSiQue | Bamboogle |
> | -------------------------- | -- | -- | --| --|
> | Naive RAG | 21.0 | 13.8 | 4.1 | 14.3
> | Joint Optimization        | 13.2 | 8.4 | 3.9 | 10.8
> | **Curriculum Learning (ours)** | **38.7**  | **40.1** | **20.9** | **39.6**
>
> Joint optimization performs poorly across all datasets -- even worse than Naive RAG -- because the model must learn the full macro-retrieval -> key-information-saving -> micro-retrieval pipeline simultaneously. This makes the optimization landscape significantly harder, prevents the model from achieving meaningful rewards, and results in inadequate training.
>
>
> In contrast, curriculum learning decomposes the training process into two tractable stages, enabling the model to first acquire macro retrieval skills before learning micro retrieval. This staged approach stabilizes optimization and yields substantially higher end-to-end performance.
>
> **Once again, we sincerely appreciate your helpful suggestions, which have enhanced the clarity, organization, and reproducibility of our paper. We hope our responses and revisions will further strengthen your confidence in our work and its evaluation.**

---

### Official Review · Reviewer_KBYP · 2025-10-26

**Soundness:** 4
**Presentation:** 3
**Contribution:** 4
**Rating:** 8
**Confidence:** 4

**Summary:**

This paper proposes Micro–Macro Retrieval (M2R), a two-level retrieve-while-generate framework designed to reduce hallucination in long-form generation. By combining macro retrieval of coarse evidence during reasoning with micro retrieval of key information during answer generation, M2R ensures that essential evidence remains proximal to output tokens. Trained with a curriculum-based reinforcement learning strategy using rule-based rewards, the method achieves significant improvements in factual consistency and robustness across multi-hop QA and long-context benchmarks compared to strong RAG baselines.

**Strengths:**

1. The idea of Micro–Macro Retrieval surprisingly natural and well-motivated — it addresses one of the most persistent issues in RAG systems (long-form hallucination) with a solution that feels both principled and minimal. The “retrieve-while-generate” framing elegantly captures how reasoning and retrieval should co-evolve.

2. The paper is very carefully written. I particularly like how the authors formalize the two retrieval levels and the transition between \<macro_tool_call>, \<key_info_save>, and \<micro_tool_call>. It feels like reading a well-designed system that could actually be implemented in production without hidden tricks.

3. I really appreciate that the system is interpretable by design: it shows that we can literally see the reasoning flow: what it retrieved, what it saved, what it reused. That’s a refreshing contrast to the black-box nature of most retrieval-augmented models. It also feels cognitively aligned with how humans solve tasks: note things down, then recall them precisely.

4. Compared with Self-RAG[1] this work feels like a thoughtful continuation rather than simple imitation. Self-RAG let the model decide when to retrieve; M2R turns that spark into a full reasoning routine. It not only detects when retrieval is needed but also explicitly manages what to keep and reuse, maintaining a long-term internal memory grounded in already verified facts. I find this progression deeply satisfying—the model isn’t merely “asking for help” anymore; it’s learning to remember what it already knows to be true. That evolution from reactive retrieval to proactive self-memory feels like a genuine step forward.

[1] Asai, Akari, et al. "Self-RAG: Learning to Retrieve, Generate, and Critique through Self-Reflection." The Twelfth International Conference on Learning Representations.

**Weaknesses:**

I think this paper has no particularly obvious weaknesses. Unlike the naïve combination of RAG with RL or GRPO, this work takes a much more principled approach—from memory to retrieval to the overall training strategy, making it a significant step forward in improving RAG performance. However, I do have a few questions that I’d like to raise briefly.

1. In your gradient computation implementation, how did you mask out the information from the retrieval part? Is the positional information handled relative to the retrieval step, or do you directly remove the masked portion?
2. When invoking retrieval, are macro-retrieval and micro-retrieval mutually exclusive, or can they be used jointly?
3. When is **key_info_save** called? In your experiments, is it triggered only when the macro retrieval is considered contextually relevant, and thus the macro information is saved? If there are identical or similar **micro_tool_calls**, does the model jointly retrieve them?
4. If the model chooses to invoke **micro**, but there is no stored memory or relevant content, does that lead to hallucination? How do you handle cases where micro-retrieval fails or retrieves incorrect results?
5. Are there situations where neither **micro** nor **macro** is used?
6. What are the overall training costs and latency characteristics? If the database is large or the query is long, does it lead to bottlenecks?
7. If provides a main figure, it will help more readers to fastly grasp your methodology.

**Questions:**

See above, I am warmly welcome to discuss further detailed on this paper.

---

> ### Author Response · Authors · 2025-11-23
> **Response to Reviewer KBYP (Part 1)**
>
> Thank you for your positive and constructive feedback. We hope our response below will further enhance your confidence in our work.
>
> ### Q1: Gradient Masking in the Retrieval Part
> > In your gradient computation implementation, how did you mask out the information from the retrieval part? Is the positional information handled relative to the retrieval step, or do you directly remove the masked portion?
>
> Thank you for the question. We adopt a direct masking strategy:
> * Retrieved spans remain in the sequence and retain their original positional indices.
> * During loss computation, we simply set their loss mask to 0, preventing any gradient from flowing through these tokens.
> * We do not modify, shift, or re-base positional encodings relative to retrieval steps.
>
> This design preserves the natural autoregressive structure while cleanly isolating gradients from the retrieval content.
>
>
> ### Q2: When invoking retrieval, are macro-retrieval and micro-retrieval mutually exclusive, or can they be used jointly?
>
> Thank you for the question. Macro retrieval and micro retrieval are mutually exclusive by design, for two reasons:
>
> * They occur in different stages of the pipeline.
> Macro retrieval is used exclusively during the \<think\> phase, while micro retrieval is restricted to the \<answer\> phase.
> Each stage serves a distinct purpose—macro retrieval gathers and stores key information, whereas micro retrieval consumes that information during answer generation.
> * There is a strict dependency order.
> Micro retrieval operates only on the distilled key information produced during macro retrieval.
> Therefore, all macro retrieval must complete before any micro retrieval can occur.
>
> ### Q3: When is key_info_save triggered, and how does micro retrieval handle similar keys?
> Thank you for the thoughtful question. We clarify both the triggering mechanism of key_info_save and the behavior of micro retrieval when similar keys exist.
>
> (1) When is key_info_save triggered?
>
> key_info_save is invoked during the \<think\> phase whenever the model identifies information it believes is relevant for answering the question.
> For example, in our case study (Table 3 in the revised paper), once the model computes the "lowest payable amount" (Line 466), it immediately calls key_info_save and stores this fact (Line 469).
>
> Importantly, this behavior is **learned**, not rule-based.
> Through GRPO, the **Key Information Correctness Reward** (Line 257) penalizes saving irrelevant or incorrect facts and rewards saving answer-aligned ones. Thus, the model learns when to checkpoint intermediate facts that matter for the final answer.
>
> (2) Is key_info_save called only after a "relevant" macro retrieval?
>
> No. Although many saved facts originate from macro-retrieved evidence, key_info_save is not gated by whether a macro retrieval has just occurred.
>
> In practice, key_info_save may occur:
> * after macro retrieval, when the retrieved passage contains a useful factual unit, or
> * during pure reasoning, when the model synthesizes previously retrieved information into a new intermediate conclusion.
>
> Empirically, 30–40% of saved facts are model-generated summaries or inferred conclusions that do not immediately follow a macro retrieval.
> This shows that key_info_save functions as a general "reasoning checkpoint", rather than a retrieval-gated action.
>
> (3) If multiple micro_tool_calls refer to identical or similar keys, does the model retrieve them jointly?
>
> Micro retrieval supports **joint execution** of all requested keys within a single batch. We do not deduplicate or merge similar keys; instead, all keys requested via micro_tool_call are retrieved and returned to the model.
>
> This design keeps micro retrieval simple, deterministic, and efficient. In practice, occurrences of multiple nearly identical keys within a single answer step are rare, so the absence of deduplication has no measurable impact on model behavior or latency.
>
> ### Q4: Handling Empty or Incorrect Micro Retrieval
> > If the model chooses to invoke micro, but there is no stored memory or relevant content, does that lead to hallucination? How do you handle cases where micro-retrieval fails or retrieves incorrect results?
>
> Thank you for the thoughtful question. Micro retrieval can encounter two distinct failure cases:
> 1. **Empty retrieval**: the model issues a micro_tool_call but the requested key is not present.
> 2. **Incorrect retrieval**: the key exists, but the stored value is inconsistent with ground truth.
>
> We describe how both cases are handled.
>
> * Empty retrieval does not cause hallucination.
>
> Empty retrieval is extremely rare. When it does occur, the tool explicitly returns: "key_name is not stored in the key-information repository."
> We instruct the model via prompting not to fabricate missing facts and to simply report the absence of retrieved information. As a result, the model avoids hallucinating unsupported content, and the answer remains conservative and grounded.

---

> ### Author Response · Authors · 2025-11-23
> **Response to Reviewer KBYP (Part 2)**
>
> * Incorrect retrieval reflects an upstream reasoning error.
>
> Incorrect retrieval arises only when the model previously saved an incorrect fact during the \<think\> phase. The micro stage then retrieves this incorrect value and uses it consistently in the final answer.
> This is not a new hallucination introduced by micro retrieval; instead, it maintains **reasoning–answer consistency** -- the final answer matches the model's earlier inference rather than inventing new content or suffering from long-context drift.
> And our **Consistency Score** in the answer reward (Line 260 in the revised paper) further reinforces this alignment and penalizes inconsistent or fabricated outputs.
>
>
> ### Q5: Are there situations where neither micro nor macro is used?
> Thank you for the question. In our implementation, such situations do not occur.
>
> The **format reward** strictly enforces the required structure: the model must perform macro retrieval during the \<think\> phase and micro retrieval during the \<answer\> phase.
>
> As a result, the policy consistently uses both retrieval stages as intended during training and inference.
>
>
>
> ### Q6: What are the overall training costs and latency characteristics? If the database is large or the query is long, does it lead to bottlenecks?
>
> Thank you for the question. We summarize training cost, inference latency, and scalability below.
>
> **Training Costs**
>
> Training is efficient because the added micro-retrieval tool is lightweight and key-information extraction remains rule-based. Using 8×A100 40GB GPUs, the two-stage curriculum framework requires:
>
> Qwen2.5-3B-Instruct: ~18 hours end-to-end training
>
> Qwen2.5-7B-Instruct: ~35 hours end-to-end training
>
> These costs are comparable to standard RL-style finetuning at similar model scales.
>
>
> **End-to-End Latency**
>
>
> | Dataset         | Avg Invocations | Latency Estimate (~100 ms each) |
> | --------------- | --------------- | ------------------------------- |
> | HotpotQA        | 5.1             | **$\approx$ 510 ms**                    |
> | 2WikiMultiHopQA | 6.2             | **$\approx$ 620 ms**                    |
> | MuSiQue         | 7.6             | **$\approx$ 760 ms**                    |
> | Bamboogle       | 4.8             | **$\approx$ 480 ms**                    |
>
>
> **Actual Inference Time (Qwen2.5-3B + SGLang, 4×A100)**
>
> | Dataset         | Avg Invocations | Avg Inference Time per Sample|
> | --------------- | --------------- | ------------------------------- |
> | HotpotQA        | 5.1             | **$\approx$  4.7 s**                |
> | 2WikiMultiHopQA | 6.2             | **$\approx$  5.2 s**                |
> | MuSiQue         | 7.6             | **$\approx$  6.8 s**                |
> | Bamboogle       | 4.8             | **$\approx$  4.6 s**                |
>
>
>
> **Scaling With Question Complexity**
>
> Using the multi-question setting from Fig. 2 (concatenating multiple HotpotQA questions):
>
> | Setting                           | Avg Invocations | Avg Inference Time per Sample |
> | --------------------------------- | --------------- | ------------------------------- |
> | 1Q (single question)              | **7.6**    |**$\approx$  6.8 s**                |
> | 2Q (two concatenated questions)   | **13.8**    |**$\approx$  14.1 s**                |
> | 3Q (three concatenated questions) | **19.7**     | **$\approx$  22.3 s**                |
>
>
> Overall, $M^2R$ exhibits linear scaling with reasoning complexity: as questions become longer or involve more hops, model invocations and latency increase proportionally, without degradation or instability.
>
> Moreover, $M^2R$ is specifically designed for long-form reasoning tasks. As shown in Fig. 2, its advantage becomes more pronounced as the input becomes longer and reasoning chains deepen. Therefore, neither a large database nor long queries produce bottlenecks in practice.
>
>
> ### Q7: If provides a main figure, it will help more readers to fastly grasp your methodology.
>
> We appreciate the valuable suggestion.
>
> In the revised manuscript, we have added a new schematic figure to visually summarize the methodology for easier comprehension. You may refer to it in the revised manuscript, or access it directly via the following anonymous link: [link to the schematic figure](https://anonymous.4open.science/r/Micro_Macro_Retrieval-E6A9/M2Rfig1.png).
>
> If you have any suggestions for improving the clarity or completeness of this figure, we would be very happy to incorporate them.
>
> **Once again, we sincerely appreciate your careful review and valuable feedback, which have substantially improved the quality and clarity of our work. We hope our response and revisions will further strengthen your confidence in our submission.**

---

> > ### Comment · Reviewer_KBYP · 2025-11-24
> > **Thanks for your response**
> >
> > Thanks for your detailed response, I would maintain my score to support your work.

---

> > > ### Author Response · Authors · 2025-11-24
> > > **Thanks to Reviewer KBYP**
> > >
> > > Dear Reviewer KBYP,
> > >
> > > Thank you very much for your thoughtful comments and positive review of our work. We sincerely appreciate your detailed feedback and strong support. Your recognition is truly encouraging and has been invaluable in strengthening this submission.
> > >
> > > Once again, thank you for your support and for dedicating your time to review our work.
> > >
> > > Best regards,
> > >
> > > Authors

---

### Official Review · Reviewer_Sm15 · 2025-10-30

**Soundness:** 2
**Presentation:** 3
**Contribution:** 3
**Rating:** 4
**Confidence:** 3

**Summary:**

This paper tackles hallucination in large language models (LLMs), especially in long-form generation where redundant contexts and extended reasoning amplify factual errors. The authors observe that factual accuracy improves when key information appears closer to the generated output, yet existing retrieval-augmented LMs (RALMs) lack mechanisms to ensure this proximity.
To address this, they propose Micro–Macro Retrieval (M2R), a retrieve-while-generate framework combining macro-level external retrieval with micro-level key information reuse from an internal repository built during reasoning. M2R directly maintains evidence proximity to outputs and reduces hallucination in long-context tasks.
Trained via curriculum-based reinforcement learning with rule-based rewards, M2R achieves consistent gains in factual accuracy and grounding across multiple benchmarks, showing good effectiveness in lengthy-context scenarios.

**Strengths:**

1. The topic is valuable, especially for mitigating hallucinations in long-form reasoning models.

2. The method is straightforward and conceptually sound.

3. The paper is well written and easy to follow.

**Weaknesses:**

1. The core methodological details are underspecified. If I understand correctly, the approach hinges on constructing and maintaining a key-information repository, and then:

- (1) For macro retrieval: per Lines 060–061, how is “the reasoning process yields answer-aligned evidence” detected, and how exactly is it inserted into the repository(i.e., what constitutes the key and the value)?

- (2) For micro retrieval: what query is used to retrieve answer-related information from the repository?

- (3) Does GRPO merely teach the LLM when/how to invoke the macro- and micro-retrieval tools, rather than optimizing the retrieved content?

2. Empirical coverage is limited. Training is conducted only on Qwen2.5-3B/7B and evaluated on four relatively simple QA datasets. How does the method perform on other model sizes/families and on more challenging reasoning benchmarks? Moreover, Figure 2 shows substantial reward oscillation and a low mean (~0.4), suggesting training stability or sufficiency may be a concern.

3. FlashRAG details and ablations require clarification, including the knowledge base size, and the effects of chunk size and retrieve number on performance. Reporting token statistics of input/output during inference would further clarify whether the approach practically alleviates long-form reasoning constraints.

**Questions:**

Please see the weaknesses,

---

> ### Author Response · Authors · 2025-11-23
> **Response to Reviewer Sm15 (Part 1)**
>
> Thank you for your valuable and constructive review. We hope that our responses to your questions will satisfactorily address your concerns.
>
> ### W1: The core methodological details are underspecified.
>
> Thank you for bringing this issue to our attention. We provide further clarification on the design of macro- and micro-retrieval in our framework.
>
> > Q1: For macro retrieval: per Lines 060–061, how is "the reasoning process yields answer-aligned evidence" detected, and how exactly is it inserted into the repository(i.e., what constitutes the key and the value)?
>
> In $M^2R$, both the detection and storage of key information are performed **directly by the model** during the \<think\> phase. As the model reasons, it decides which intermediate facts should be written into the key-information repository. To ensure that only relevant evidence is stored, we apply the **Key Information Correctness Reward** (Line 257 in the revised paper), which penalizes irrelevant or incorrect entries and encourages saving answer-aligned facts.
>
>
> For example, in our case study (Table 3), once the model calculates the *"lowest payable amount"* (Line 466), it immediately writes this fact into the repository (Line 469). Thus, each stored item is a key–value pair:
> * **Key**: a concise semantic label summarizing the fact
>   e.g., "finalPayableAmount"
> * **Value**: the atomic fact produced during reasoning
>   e.g., "180"
>
> This mechanism ensures that the repository contains compact, model-generated, answer-relevant factual units.
>
>
> > Q2: For micro retrieval: what query is used to retrieve answer-related information from the repository?
>
>
> The key-information repository is implemented as a dictionary-like structure holding all key–value pairs generated during the \<think\> phase.
>
> During the \<answer\> phase, the model issues a \<micro_tool_call\> with a "query" field set to exactly the key of the fact it wants to retrieve:
>
> \<micro_tool_call\>
> "query": KEY_NAME
> \</micro_tool_call\>
>
> For instance, in Table 3, Line 474, the model retrieves:
> "query": "finalPayableAmount"
>
> The returned value is then used to ground the final answer.
> This design ensures precise and deterministic retrieval of previously generated evidence.
>
>
>
> > Q3: Does GRPO merely teach the LLM when/how to invoke the macro- and micro-retrieval tools, rather than optimizing the retrieved content?
>
> Your understanding is correct. GRPO in $M^2R$ does not optimize the retrieval module itself. Instead, GRPO optimizes the model’s generation behavior, specifically:
> * when to trigger macro- or micro-retrieval
> * how to compose and sequence the tool calls
> * what key information should be written during reasoning
> * how retrieved information should be incorporated into the answer
>
> Because retrieval is not modified, $M^2R$ is agnostic to the underlying retrieval system and can be combined with stronger retrieval enhancements in future work without conflict.
>
> We have added the above clarifications to the revised paper (Lines 83 and 216–222) and marked all corresponding changes in blue for easy reference.

---

> ### Author Response · Authors · 2025-11-23
> **Response to Reviewer Sm15 (Part 2)**
>
> ### W2: Empirical coverage is limited.
> > Training is conducted only on Qwen2.5-3B/7B and evaluated on four relatively simple QA datasets. How does the method perform on other model sizes/families and on more challenging reasoning benchmarks? Moreover, Figure 2 shows substantial reward oscillation and a low mean (~0.4), suggesting training stability or sufficiency may be a concern.
>
> Thank you for the thoughtful question. We provide additional experiments and address concerns about training stability and sufficiency.
>
> **Experiments on Different Model Families**
>
> To assess generalizability, we evaluate $M^2R$ on Llama-3.1-8B-Instruct and Mistral-7B-Instruct. Exact Match results are shown below.
>
> **Results on Llama-3.1-8B-Instruct**
> | Method           | HotpotQA | 2Wiki    | MuSiQue  | Bamboogle |
> | ------ | ------- | ------ | ------ | ------ |
> | Naive Generation | 18.1     | 24.7     | 4.4      | 11.6       |
> | Naive RAG        | 34.2     | 31.3     | 9.8      | 20.9      |
> | COFT             | 38.4     | 39.5     | 15.8     | 32.5      |
> | SURE             | 39.3     | 42.0     | 18.7     | 37.8      |
> | ReSearch         | 42.2     | 45.8     | 20.9     | **43.1**      |
> | **$M^2R$ (ours)**   | **43.0** | **47.2** | **22.1** | 42.9  |
>
>
> **Results on Mistral-7B-Instruct**
> | Method           | HotpotQA | 2Wiki    | MuSiQue  | Bamboogle |
> | --------- | -------- | -------- | -------- | --------- |
> | Naive Generation | 21.9     | 28.8     | 7.7      | 12.5      |
> | Naive RAG        | 34.5     | 31.2     | 11.3      | 25.4      |
> | COFT             | 43.5     | 45.5     | 18.5     | 40.3      |
> | SURE             | 42.6     | 47.7     | 21.2     | 42.8      |
> | ReSearch         | 45.0     | 49.1     | 23.7     | 45.5      |
> | **$M^2R$ (ours)**   | **45.6** | **50.0** | **25.5** | **46.0**  |
>
>
> Across both model families, $M^2R$ outperforms ReSearch in 7 out of 8 cases, highlighting its strong cross-model robustness.
>
> **Experiments on More Challenging Reasoning Benchmarks**
>
> To assess longer reasoning chains and more complex tool-use dynamics, we extend the evaluation by requiring the model to answer multiple questions in a single input (3Q or 5Q variants). Results on Qwen2.5-7B-Instruct are shown below.
>
> | Method           | HotpotQA-3Q | 2Wiki-3Q | MuSiQue-3Q | Bamboogle-3Q |
> | --------- | ------- | -------- | ---------- | ------------ |
> | Naive Generation | 13.1         | 17.5      | 4.3        | 7.1          |
> | Naive RAG        | 20.6        | 16.9     | 5.1        | 15.2          |
> | COFT             | 24.5        | 27.2     | 12.6        | 22.0         |
> | SURE             | 28.3        | 31.5     | 11.3        | 25.5         |
> | ReSearch         | 30.2        | 33.9     | 14.2        | 28.0         |
> | **$M^2R$ (ours)**   | **32.0**    | **35.8** | **17.9**   | **30.6**     |
>
> | Method           | HotpotQA-5Q | 2Wiki-5Q | MuSiQue-5Q | Bamboogle-5Q |
> | -------- | ----------- | -------- | --------- | --------- |
> | Naive Generation | 5.5         | 4.5      | 0.7        | 1.8          |
> | Naive RAG        | 8.2         | 7.7      | 2.3        | 4.5          |
> | COFT             | 9.5         | 11.1     | 3.5        | 9.5          |
> | SURE             | 13.1        | 14.8     | 4.8        | 10.7         |
> | ReSearch         | 13.9        | 17.0     | 5.7        | 12.8         |
> | **$M^2R$ (ours)**   | **15.4**    | **18.5** | **8.4**    | **14.9**     |
>
> The results show that $M^2R$ achieves significant improvements under extended reasoning chains, effectively addressing the lost-in-lengthy-contexts issue and demonstrating strong performance in long-form reasoning tasks.
>
> **Training Stability and Sufficiency**
>
> We ensure training stability from two complementary aspects:
>
> **(1) Stabilizing GRPO training**
> * We prioritize using a large ROLLOUT_N across all model sizes to ensure sufficient reward samples for each update.
> * We adopt dynamic sampling, discarding zero-reward trajectories to keep updates meaningful.
>
> **(2) Reducing learning difficulty through curriculum**
>
> Training both retrieval stages from scratch yields extremely sparse rewards because many generations fail the Format Reward check. Our curriculum learning design decomposes the learning process into manageable stages, significantly improving stability and convergence.
>
> For training sufficiency, a low-scale reward is expected and does not indicate a failure to learn:
> 1. A reward > 0.1 already implies correct macro/micro tool-use behavior, as it must pass the Format Reward gate.
> 2. The answer reward itself underestimates accuracy:
> We follow ReSearch and compute the answer reward using F1 score. However, F1 score is brittle when the answer surface forms differ even slightly, making semantically correct answers score 0 (e.g., "$25.20" vs. "25.2").
>
> As a result, the actual correctness of the policy model is higher than what the F1-based reward suggests. This also explains why LLM-as-a-Judge scores > Exact Match, as shown in Table 1.

---

> ### Author Response · Authors · 2025-11-23
> **Response to Reviewer Sm15 (Part 3)**
>
> We intentionally keep the same scoring setup to ensure **fair comparison** -- i.e., improvements come from the proposed framework rather than from reward engineering.
>
>
>
> We agree that there is still significant potential for advancing reward design. For instance, further considering the relationships among the key-information repository, model outputs, and ground-truth answers, as well as refining the reward scoring scheme, could further enhance the effectiveness of $M^2R$. We view this as a promising direction for future research.
>
>
> ### W3: FlashRAG details and ablations require clarification.
> > FlashRAG details and ablations require clarification, including the knowledge base size, and the effects of chunk size and retrieve number on performance. Reporting token statistics of input/output during inference would further clarify whether the approach practically alleviates long-form reasoning constraints.
>
> Thank you for the helpful suggestion. We clarify the FlashRAG configuration and provide additional ablations and token statistics below.
>
>
> **FlashRAG Configuration**
>
> To ensure a fair and retrieval-agnostic comparison, we strictly adopt the official ReSearch / FlashRAG setup without modification:
> * Knowledge Base: FlashRAG’s December 2018 Wikipedia snapshot
> * Chunk Size: Default ~100-word passages (default)
> * Retriever Model: E5-base-v2 dense retriever
> * Retrieval hyperparameters: topk = 5, batch_size = 256, query_max_length = 128 (FlashRAG and ReSearch defaults)
>
> This ensures that the observed improvements come from $M^2R$’s generation mechanism, rather than from tuning the retrieval component.
>
> **Ablations on Retrieval Settings**
>
> We ablate two key FlashRAG parameters -- retrieve-top-k and chunk size -- on 2Wiki using Qwen2.5-3B:
>
> | Retrieve-Top-k  | Naive RAG | ReSearch | **$M^2R$ (ours)** |
> | --------------- | --------- | -------- | -------------- |
> | **3**           | 13.5      | 37.2     | **38.3**       |
> | **5 (default)** | 13.8      | 38.9     | **40.1**       |
> | **8**           | 13.6      | 38.0     | **39.4**       |
>
> | Chunk Size        | Naive RAG | ReSearch | **$M^2R$ (ours)** |
> | ----------------- | --------- | -------- | -------------- |
> | **50**            | 13.4      | 38.1     | **39.4**       |
> | **100 (default)** | 13.8      | 38.9     | **40.1**       |
> | **150**           | 13.9      | 38.4     | **39.7**       |
>
> Across all settings, $M^2R$ consistently outperforms ReSearch, indicating strong robustness to retrieval hyperparameters.
>
>
>
> **Token Statistics During Inference**
>
> To evaluate whether $M^2R$ alleviates long-form reasoning constraints, we report input and output token statistics across datasets. Note that "Avg. Input Tokens" refers only to the question tokens, while "Avg. Output Tokens" include both the reasoning chains and the final answer, excluding all retrieval passages.
>
>
> | Dataset       | Avg. Input Tokens | Avg. Output Tokens (ReSearch) | Avg. Output Tokens ($M^2R$) |
> | ------------- | ----------------- | ------------------------------- | -------------------------- |
> | **HotpotQA**  |        25         | 416 | 432
> | **MuSiQue**   |        31            | 483  | 505
> | **2Wiki**     |        37            | 440 | 478
> | **Bamboogle** |  21 | 376 | 389
>
> The input token lengths are identical across methods. For output tokens, $M^2R$ generates slightly longer outputs due to the micro-retrieval and key-information saving steps, but these operations inject compact, answer-aligned facts near the model's generation point, which leads to performance gains.
>
> We have included these additional experimental results in the revised manuscript, highlighted in blue (Lines 492–522, Lines 902-1128).
>
> **Once again, we sincerely appreciate your time and thoughtful feedback. We have provided a detailed response to your comments and thoroughly incorporated your suggestions in revising the paper, which have significantly strengthened the quality and completeness of our work. We hope our response and revisions will prompt you to reassess our submission.**

---

### Official Review · Reviewer_Jaoq · 2025-11-01

**Soundness:** 2
**Presentation:** 3
**Contribution:** 3
**Rating:** 4
**Confidence:** 3

**Summary:**

A "retrieve-while-generate" framework that performs macro retrieval (external sources during reasoning) and micro retrieval (from key-information repository during answer generation) to ensure evidence proximity and reduce hallucination.

**Strengths:**

1. Strong Empirical MotivationLost in the Middle Phenomenon (Liu et al. 2023):

GPT-3.5 accuracy drops from 75% → 55% when answer-bearing evidence moves from context start to middle (Figure 3)
Empirically validated across multiple models and tasks
Theoretical Grounding (Appendix B):

Analyzes RoPE positional encoding: attention ∝ q^T R_{θ,m-n} k
High-frequency components cancel at large distances Δ
Formal claim: "Evidence contribution decreases monotonically with distance"
This foundation is solid — problem is real and well-documented.

2. Explicit Key Information ManagementKey-Value Repository Design:

Advantages:

- Explicitly separates "what to remember" from "how to answer"
- Forces model to extract atomic facts rather than rely on context attention
- Reduces cognitive load during answer generation
This is cleaner than implicit reasoning traces (e.g., ReSearch where key info is buried in <think> text).

**Weaknesses:**

1. "Retrieve-While-Generate" might be Misleading

What the Paper Claims:

"M²R is the first framework to introduce a retrieve-while-generate paradigm during the answer phase."
This suggests a novel generation mechanism — e.g., retrieval happening during the forward pass.What Actually Happens:Multi-Turn Generation with Tool Calls:

This is identical to:

OpenAI's function calling
Anthropic's tool use
ReAct (Yao et al. 2022)
Self-RAG (Asai et al. 2023) — which also retrieves during answer generation!
M²R requires 5-10 sequential model invocations per query.

2. Cost Analysis Completely Absent

**Questions:**

1. How many model invocations does M²R require per query on average?

Please report separately for: (a) think phase, (b) answer phase, (c) total
Break down by dataset (HotpotQA, MuSiQue, etc.)
What is the range (min/max)?



2. What is the end-to-end latency in realistic deployment?

Table 2 shows batch inference time (0.67s), but what about non-batched API calls?
Assuming 100ms per forward pass: how long does a typical query take?
How does this scale with question complexity?

---

> ### Author Response · Authors · 2025-11-23
> **Response to Reviewer Jaoq (Part 1)**
>
> Thank you for your valuable and constructive comments and for recognizing the strong empirical and theoretical motivation behind $M^2R$. We address your concerns in detail below.
>
> ### W1: Differences with Existing "Retrieve-While-Generate" Methods.
> > What the Paper Claims: "$M^2R$ is the first framework to introduce a retrieve-while-generate paradigm during the answer phase." This suggests a novel generation mechanism — e.g., retrieval happening during the forward pass.What Actually Happens:Multi-Turn Generation with Tool Calls: This is identical to: OpenAI's function calling Anthropic's tool use ReAct (Yao et al. 2022) Self-RAG (Asai et al. 2023) — which also retrieves during answer generation! M²R requires 5-10 sequential model invocations per query.
>
> We appreciate the reviewer’s attention to this issue and acknowledge that our previous wording may not have been appropriate. We would like to clarify that, while prior work indeed performs multi-turn retrieval during generation, **our $M^2R$ differs in two essential aspects that are not present in these methods.**
>
>
>   * **Difference 1. During the answering phase, $M^2R$ retrieves internal, model-generated key information (micro retrieval), rather than retrieving external documents.**
>
> Existing RAG or tool-use frameworks retrieve **only external documents**, and and do not utilize **model-generated intermediate reasoning** during answer decoding.
>
> In contrast, $M^2R$ constructs an internal key-information repository during the \<think\> (reasoning) phase and performs micro retrieval exclusively over these model-generated atomic facts during the answering phase. Such an approach is absent in previous RAG methods, and current tool-use systems also lack mechanisms for retrieving model-generated evidence.
>
> To the best of our knowledge, **$M^2R$ is the first method to retrieve model-generated key information during the answer stage**, setting it apart from approaches that retrieve external evidence.
>
> * **Difference 2. $M^2R$ explicitly enforces evidence proximity during answer generation — absent in all prior methods.**
>
>
> Although prior frameworks may retrieve during answer generation, **none of them control the proximity of retrieved evidence to the output tokens.**
>
> Motivated by the lost-in-the-middle phenomenon, $M^2R$’s micro retrieval is explicitly designed to place the key evidence as close as possible to the output tokens, thereby reducing factual drift in long-form generation.
>
>
> To avoid misunderstanding, we have revised the wording in our paper as follows:
>
>
> >>"$M^2R$ introduces a new retrieve-while-generate mechanism during the answer phase, where retrieval is performed over model-generated key information, and answer generation is constrained by enforcing proximity between the retrieved evidence and the generated tokens."
>
>
> We have incorporated the above distinctions into the Related Work section to more clearly position our contributions, and have added a new schematic figure to visually summarize the methodology for easier comprehension. You may refer to it in the revised manuscript, or access it directly via the following anonymous link: [link to the schematic figure](https://anonymous.4open.science/r/Micro_Macro_Retrieval-E6A9/M2Rfig1.png).

---

> ### Author Response · Authors · 2025-11-23
> **Response to Reviewer Jaoq (Part 2)**
>
> ### W2 & Q1 & Q2. Cost Analysis Absent
>
> Thank you for your valuable suggestion. Below we provide detailed statistics on model invocations and end-to-end latency.
>
>
> **Number of Model Invocations per Query**
>
> We evaluate the average number of model calls for both the \<think\> (macro retrieval) and \<answer\> (micro retrieval) phases using Qwen2.5-3B-Instruct.
>
> | Dataset         | Think Phase | Answer Phase | Total   | Min | Max |
> | --------------- | ----------- | ------------ | ------- | --- | --- |
> | HotpotQA        |  3.7 | 1.4 | 5.1 | 3 | 6
> | 2Wiki |  4.5 | 1.7 | 6.2 | 3 | 10
> | MuSiQue         |  5.7 | 1.9 | 7.6 | 4 | 9
> | Bamboogle       |  3.5 | 1.3 | 4.8 | 2 | 6
>
> The resutls show that most invocations come from the \<think\> phase -- a cost already required by all multi-turn tool-based RAG frameworks. The additional overhead from $M^2R$ is only 1–2 extra calls from micro retrieval, corresponding to roughly a 20–30% relative increase.
>
> Micro retrieval itself is extremely cheap: it performs a simple rule-based lookup over a very small local repository, so the added runtime is minimal.
>
>
> **End-to-End Latency Estimates**
>
> We would like to clarify that our Table 2 does not report any inference-time or latency measurements (such as the "0.67s" mentioned in the comment). Table 2 only reports accuracy metrics.
>
> Following the reviewer’s request, we report approximate end-to-end latency assuming ~100 ms per forward pass:
>
> | Dataset         | Avg Invocations | Latency Estimate (~100 ms each) |
> | --------------- | --------------- | ------------------------------- |
> | HotpotQA        | 5.1             | **$\approx$ 510 ms**                    |
> | 2WikiMultiHopQA | 6.2             | **$\approx$ 620 ms**                    |
> | MuSiQue         | 7.6             | **$\approx$ 760 ms**                    |
> | Bamboogle       | 4.8             | **$\approx$ 480 ms**                    |
>
>
> **Actual Inference Time**
>
> We also benchmark real latency including all tool-calling and retrieval overhead, using Qwen2.5-3B with SGLang on a 4×A100 40GB machine.
>
> | Dataset         | Avg Invocations | Avg Inference Time per Sample|
> | --------------- | --------------- | ------------------------------- |
> | HotpotQA        | 5.1             | **$\approx$  4.7 s**                |
> | 2WikiMultiHopQA | 6.2             | **$\approx$  5.2 s**                |
> | MuSiQue         | 7.6             | **$\approx$  6.8 s**                |
> | Bamboogle       | 4.8             | **$\approx$  4.6 s**                |
>
>
> **Scaling with Input Complexity**
>
> We further evaluate latency under multi-question reasoning (concatenating 2–3 HotpotQA questions), using the same hardware setup:
>
> | Setting                           | Avg Invocations | Avg Inference Time per Sample |
> | --------------------------------- | --------------- | ------------------------------- |
> | 1Q (single question)              | **7.6**    |**$\approx$  6.8 s**                |
> | 2Q (two concatenated questions)   | **13.8**    |**$\approx$  14.1 s**                |
> | 3Q (three concatenated questions) | **19.7**     | **$\approx$  22.3 s**                |
>
> Latency grows approximately linearly with question complexity, which is expected in multi-turn reasoning.
>
> We have included these invocation statistics and latency measurements in the revised manuscript (Lines 492–522, Lines 902-1128).
>
> **Once again, we sincerely appreciate your time and constructive feedback. We hope that our additional analysis and clarifications have satisfactorily addressed your concerns and encourage a reassessment of our work.**

---

### Author Response · Authors · 2025-11-23

Dear Reviewers,

We sincerely appreciate the reviewers’ thorough reading of our submission and the constructive insights provided. Over the past several days, we have carefully revised the manuscript and prepared a detailed point-by-point response. The key revisions and additions are summarized below:


- **Providing additional experimental results** (`Reviewers Jaoq, Sm15, KBYP and vzvj`), including latency and time-cost analysis, retrieval ablations, and evaluations on additional model families including Llama-3.1-8B-Instruct and Mistral-7B-Instruct and more challenging benchmarks such as extended multi-question variants of the original datasets.
- **Clarifying the methodological formulation and technical design details of the proposed approach** (`Reviewers Jaoq, Sm15, KBYP and vzvj`).
- **Highlighting the differences between our approach and existing multi-turn tool-call RAG frameworks** (`Reviewer Jaoq`).

We address each reviewer’s comments individually and have incorporated all revisions and additional experimental results into the revised manuscript.

Once again, we would like to express our gratitude for the reviewers’ insightful feedback, which has substantially improved the clarity, comprehensiveness, and completeness of our work. We have made every effort to address all comments thoroughly.

Thank you for your time and consideration.

Best regards,

The Authors

---

### Meta-Review · Area_Chair_are2 · 2025-12-22

**Summary:**

The paper proposes Micro-Macro Retrieval (M²R) to reduce hallucination by enforcing evidence proximity. Reviewers praised the motivation and interpretability but raised significant concerns regarding the limited model diversity (originally only Qwen) and the lack of cost analysis (latency/invocations). The rebuttal was substantive: the authors conducted new experiments on Llama-3.1 and Mistral-7B (addressing the diversity concern) and provided detailed latency and storage statistics (addressing the cost concern). While only one reviewer explicitly confirmed their score post-rebuttal, the objective provision of all requested data supports acceptance.

**Reviewer Concerns:**

**Addressed:**

- Limited Model Family (Reviewers Sm15, vzvj): The authors convincingly addressed this by adding Llama-3.1 and Mistral-7B benchmarks, showing the method generalizes beyond the Qwen family.

- Missing Cost Data (Reviewers Jaoq, vzvj): The authors provided the requested tables for latency and storage.

**Outstanding:**

- High Inference Latency (Reviewer Jaoq): While the authors provided the data, the data itself confirms a significant weakness. The method requires 5–8 model invocations per query, resulting in an end-to-end latency of ~4.7 to 6.8 seconds (on A100s). For many real-world applications, this latency is prohibitive compared to standard RAG. The rebuttal clarifies the cost but does not mitigate it.

- Incremental Novelty (Reviewer Jaoq): The reviewer challenged the "retrieve-while-generate" framing, arguing it is essentially standard multi-turn tool use (like ReAct or Self-RAG). The authors argued that retrieving internal states is novel. However, critically, this can still be viewed as a specific engineering instance of a scratchpad/memory-augmented agent rather than a fundamental paradigm shift. The distinction between "Micro Retrieval" and simply "reading from a structured context/scratchpad" is thin.

- Training Complexity (Reviewer Sm15): The method relies on a complex pipeline involving Curriculum Learning + GRPO (Reinforcement Learning). Reviewer Sm15 noted low reward stability. While the authors claimed this is "expected," reproducing this complex training pipeline (vs. standard SFT) will be challenging for the community. The dependency on carefully tuned rule-based rewards for "Key Information Correctness" introduces fragility.

**Reviewer Scores:**

Reviewer KBYP: 8 (Confirmed).

Reviewer vzvj: 6 (Likely maintained). While the data was provided, the confirmation of high latency might prevent a score increase to 8.

Reviewer Jaoq: 4 (Likely maintained). The reviewer was critical of the framing and cost. The rebuttal confirmed the high cost (linear scaling with complexity). The reviewer might acknowledge the effort but remain skeptical of the practicality.

Reviewer Sm15: 4 -> Borderline accept. The added experiments help, but the concern about the method being "underspecified" or complex to train likely remains, given the reliance on RL dynamics.

---

### Decision · Program_Chairs · 2026-01-26

Accept (Poster)